# A Scalable Global Optimization Algorithm For Constrained Clustering

## Abstract

Constrained clustering leverages limited domain knowledge to improve cluster-
ing performance and interpretability, but incorporating pairwise *must-link* and
*cannot-link* constraints is an NP-hard challenge, making global optimization in-
tractable. Existing mixed-integer optimization methods are confined to small-scale
datasets, limiting their utility. We propose Sample-Driven Constrained Group-
Based Branch-and-Bound (SDC-GBB), a decomposable branch-and-bound (BB)
framework that collapses must-linked samples into centroid-based pseudo-samples
and prunes cannot-link through geometric rules, while preserving convergence and
guaranteeing global optimality. By integrating grouped-sample Lagrangian de-
composition and geometric elimination rules for efficient lower and upper bounds,
the algorithm attains highly scalable pairwise k-Means constrained clustering via
parallelism. Experimental results show that our approach handles datasets with
200,000 samples with cannot-link constraints and 1,500,000 samples with must-
link constraints, which is 200 - 1500 times larger than the current state-of-the-art
under comparable constraint settings, while reaching an optimality gap of $\leq 3\%$.
In providing deterministic global guarantees, our method also avoids the search
failures that off-the-shelf heuristics often encounter on large datasets.

## 1 Introduction

Clustering is a core task in unsupervised learning, widely used in pattern recognition, data mining,
and computer vision (Jain, 2010; Jain et al., 1999; Rao, 1971). However, purely unsupervised
methods often overlook domain-specific requirements, motivating the integration of prior knowledge.
Constrained clustering addresses this by incorporating guidance—typically as *must-link* or *cannot-link*
constraints—to improve clustering alignment with real-world applications (Basu et al., 2008; Brieden
et al., 2017; Tian et al., 2021). These methods have been applied to facility location, genomics, image
segmentation, and text analysis (Yang et al., 2022; Zhang et al., 2022; Pelegrín, 2023).

Minimizing the within-cluster sum-of-squares, known as the Minimum Sum-of-Squares Criterion
(MSSC) (Späth, 1980), is a key goal in clustering. To handle MSSC with constraints, many methods
adapt unsupervised algorithms using penalties or assignment modifications for must-link (ML) and
cannot-link (CL) constraints. For instance, constrained $k$-means Wagstaff et al. (2001) assigns
points greedily to the nearest feasible center, but lacks optimality guarantees and may miss feasible
solutions due to MSSC's non-convexity and initialization sensitivity (Xu & Lange, 2019; Piccialli
et al., 2022a). Heuristic variants address this by refining assignments (ICOP-$k$-means (Tan et al.,
2010; Rutayisire et al., 2011)), leveraging assistant centroids (MLC-$k$-means (Huang et al., 2008)),
or delaying constraint enforcement (Nghiem et al., 2020). Soft constraint methods like PCKmeans
(Basu et al., 2004; Davidson & Ravi, 2005) integrate penalties directly into the objective.

Although heuristic algorithms are scalable and easy to implement, they lack global optimality
guarantees and may fail to find feasible solutions. Exact methods for unconstrained MSSC have
been well-studied (Hua et al., 2021; Piccialli et al., 2022b; Aloise et al., 2009), but extending them to
constrained settings is difficult due to the added combinatorial complexity of constraint satisfaction.
As a result, exact approaches remain limited to small datasets. For instance, Xia (2009) extended
the global optimization framework of Peng & Xia (2005) to handle constraints but scaled only to 25
points (Aloise et al., 2012a). Later, column-generation methods (Aloise et al., 2012b; Babaki et al.,
2014) supported slightly larger instances (under 200 points) with limited constraints.

Constraint programming approaches (Dao et al., 2013; 2015; 2017) offer flexibility in incorporating various constraint forms but generally do not scale beyond a few hundred points. Mixed-Integer Programming (MIP) formulations have also been explored to handle additional cluster-level or instance-level constraints in MSSC. For example, (Tang et al., 2020) proposed an iterative scheme that reformulates a size-constrained MSSC problem into a mixed-integer linear program, leveraging the unimodularity of certain constraint matrices to reduce computational complexity. Similarly, (Liberti & Manca, 2022) examined several side-constrained MSSC models cast as Mixed-Integer Nonlinear Programs, some featuring convex relaxations that enable global optimization techniques. While these MIP-based approaches provide a powerful and flexible framework for ensuring feasibility under various constraint types, their applicability remains limited by high computational overhead, restricting them to relatively small or moderate-sized datasets.

Branch-and-bound methods have also been specialized for constrained MSSC. (Guns et al., 2016) proposed the Constraint Programming Repetitive Branch-and-Bound Algorithm (CPRBBA), which augments Brusco's repetitive branch-and-bound procedure (Brusco, 2006) with a constraint programming solver to compute tight lower and upper bounds on subsets of objects of increasing size. This approach, while effective on small instances, remains limited to fewer than 200 data points. (Piccialli et al., 2022a) developed the PC-SOS-SDP algorithm, which integrates *must-link* and *cannot-link* constraints into a semidefinite programming framework, scaling to a few thousand data points but not beyond (Baumann & Hochbaum, 2024). These exact methods do not generally account for soft constraints and remain computationally expensive for larger datasets. Nonetheless, continuing progress in algorithmic design and hardware (Bertsimas & Dunn, 2017) has widened the scope of exact methods for constrained clustering.

**Our Contributions** In this paper, we propose a scalable deterministic global optimization algorithm for the minimum sum-of-squared clustering (MSSC) task *with pairwise ML and CL constraints*. We introduce a centroid-based pseudosample formulation for must-link subsets, leveraging the combined information of each group to maintain the exact global minimum while reducing problem complexity. We devise geometric sample-determination rules that eliminate cannot-links, which specify whether points must not be placed into the same clusters before enumeration. We design a branch-and-bound algorithm that branches only on the cluster-center variables. This avoids combinatorial branching on sample-to-cluster assignments, thus achieving a globally $\epsilon$-optimal solution even for large-scale datasets. Our analysis proves convergence under exhaustive subdivisions of the feasible region for the center variables.

**Capability For More than One Hundred Thousand Scale Problems** We present an open-source implementation in `Julia` that solves constrained MSSC instances of up to 1,500,000 samples for the ML case and 200,000 samples for CL case with optimality guarantee or very low optimality gaps. This corresponds to 1500-fold and 200-fold increases in scale, respectively, over the current exact state-of-the-art (Piccialli et al., 2022a). This framework thus enables deterministic global clustering solutions for large-scale datasets previously considered intractable.

## 2 MIXED-INTEGER PROGRAMMING FOR PAIRWISE-CONSTRAINED $k$-MEANS

Given a dataset $X = \{x_1, \ldots, x_S\} \subset \mathbb{R}^m$ with $S$ samples and $m$ attributes, the semi-supervised MSSC task with pairwise constraints seeks a set of $k$ clusters that minimizes the Sum of Squared Errors (SSE) subject to must-link (ML) and cannot-link (CL) requirements:

$$\min_b \sum_{s \in \mathcal{S}} \sum_{k \in \mathcal{K}} b_{s,k} \|x_s - \mu_k\|_2^2 \tag{1a}$$

$$\text{s.t. } b_{s,k} = b_{s',k}, \qquad \forall (s, s') \in \mathcal{T}_{ml}, \ k \in \mathcal{K}, \tag{1b}$$

$$b_{s,k} + b_{s',k} \leq 1, \qquad \forall (s, s') \in \mathcal{T}_{cl}, \ k \in \mathcal{K}, \tag{1c}$$

$$b_{s,k} \in \{0, 1\}, \qquad \forall s \in \mathcal{S}, \ k \in \mathcal{K}. \tag{1d}$$

$$\sum_{k \in \mathcal{K}} b_{s,k} = 1 \tag{1e}$$

where $s \in \mathcal{S} := \{1, \cdots, S\}$ is the data sample set, $k \in \mathcal{K} = \{1, \cdots, K\}$ is the cluster set, $\mu := [\mu_1, \cdots, \mu_K]$, where $\mu_k \in \mathbb{R}^m$ represents the center of each cluster, $b_{s,k} \in \{0, 1\}$ is equal to

1 if $x_s$ belongs to the $k$-th clusters and 0 otherwise. $\mathcal{T}_{ml} \subseteq \mathcal{S} \times \mathcal{S}$ and $\mathcal{T}_{cl} \subseteq \mathcal{S} \times \mathcal{S}$ are the sets of tuples indicating whether samples must or must not reside in the same cluster respectively.

The MSSC with pairwise constraints (Problem 1) can be reformulated as an SSE optimization problem of the following form:

$$\min_{\mu,d,b} \sum_{s \in \mathcal{S}} d_{s,*} \tag{2a}$$

$$\text{s.t.} \quad -N(1-b_{s,k}) \leq d_{s,*} - d_{s,k} \leq N(1-b_{s,k}) \tag{2b}$$

$$d_{s,k} \geq \|x_s - \mu_k\|_2^2 \quad \forall s \in \mathcal{S}, \forall k \in \mathcal{K} \tag{2c}$$

$$\text{Constraints 1b- 1e} \tag{2d}$$

Here $d_{s,k}$ is the distance between $x_s$ and $\mu_k$, $d_{s,*}$ is the distance from $x_s$ to its assigned centroid, and $N$ is a big-$M$ constant. Define $d_s = [d_{s,1}, \ldots, d_{s,K}, d_{s,*}]$, $d = [d_1, \ldots, d_S]$, $b_s = [b_{s,1}, \ldots, b_{s,K}]$, $b = [b_1, \ldots, b_S]$. Constraint (2b) links $d_{s,*}$ and $d_{s,k}$ when $b_{s,k} = 1$. Problem 2 is a mixed-integer second-order cone program (MISOCP) admits a two-stage extensive form (see Appendix A). While off-the-shelf solvers like Gurobi (Gurobi, 2024) and CPLEX (Cplex, 2022) can handle small instances, they become intractable even at moderate sample sizes (e.g., $S = 800$) (Piccialli et al., 2022a).

## 3 REDUCED-SPACE BRANCH-AND-BOUND ALGORITHM

Reduced-space branch-and-bound frameworks have demonstrated significant scalability gains by partitioning only the centroid search space (Cao & Zavala, 2019). We tailor this scheme to the pairwise-constrained clustering problem by integrating geometric probing rules derived from the MISOCP formulation in Sec. 2 to tighten both lower and upper bounds. In particular, we exploit the implicit inequality that any subregion whose lower bound exceeds the current best upper bound can be discarded outright.

### 3.1 GEOMETRIC SAMPLE DETERMINATION RULES

We first observe that every feasible clustering is subject to two straightforward geometric bounds relative to the incumbent solution. Let $\rho = \max_{s \in \mathcal{S}} \|\mathbf{x}_s - \boldsymbol{\mu}_{k(s)}^{\text{best}}\|_2^2$ be the worst-case squared distance between each sample and the centroid to which it is assigned in the current best solution. Thus, $\rho$ represents the maximum per-sample contribution to the current incumbent cost and is used as a per-sample upper bound. Then, for any region $M_k$ (an axis–aligned box in $\mathbb{R}^m$) containing the true optimal $\mu_k$, we can compute the minimal and maximal possible squared distances:

$$d_{\min}(\mathbf{x}_s, M_k) = \min_{\mu \in M_k} \|\mathbf{x}_s - \mu\|_2^2, \quad d_{\max}(\mathbf{x}_s, M_k) = \max_{\mu \in M_k} \|\mathbf{x}_s - \mu\|_2^2.$$

Because $\rho$ is an upper bound on the true assignment cost, any candidate pair $(s, k)$ with $d_{\min}(\mathbf{x}_s, M_k) > \rho$ can never be optimal.

**Lemma 3.1** (Early–elimination). *For any sample $s \in \mathcal{S}$ and any cluster region $M_k$, $d_{\min}(\mathbf{x}_s, M_k) > \rho \implies b_{s,k} = 0$ in every optimal solution with objective value not larger than the incumbent.*

*Proof.* By definition $d_{\min}(\mathbf{x}_s, M_k) = \min_{\mu \in M_k} \|\mathbf{x}_s - \mu\|_2^2$. If $d_{\min} > \rho$, then for every $\mu \in M_k$ one has $\|\mathbf{x}_s - \mu\|_2^2 > \rho$. Since $\rho$ is an upper bound on the per–sample cost in the incumbent, assigning $\mathbf{x}_s$ to cluster $k$ would yield a contradiction. Hence $b_{s,k} = 0$ in any cluster whose overall cost does not exceed the incumbent cost. $\square$

A complementary rule arises from comparing the worst-case assignment cost in one region to the best-case cost in the others.

**Lemma 3.2** (Forced assignment). *Fix a sample $s$ and let $k^+ \in \mathcal{K}$ satisfy $d_{\max}(\mathbf{x}_s, M_{k^+}) < \min_{k \neq k^+} d_{\min}(\mathbf{x}_s, M_k)$. Then $b_{s,k^+} = 1$ in every optimal solution.*

*Proof.* For any $\mu^+ \in M_{k^+}$ and any $\mu \in M_k$ with $k \neq k^+$ we have $\|\mathbf{x}_s - \mu^+\|_2^2 \leq d_{\max}(\mathbf{x}_s, M_{k^+}) < d_{\min}(\mathbf{x}_s, M_k) \leq \|\mathbf{x}_s - \mu\|_2^2$. Thus the distance from $\mathbf{x}_s$ to every center in $M_{k^+}$ is strictly smaller than the distance to any center in the remaining regions, implying that the unique cost–minimising assignment is $b_{s,k^+} = 1$. $\square$

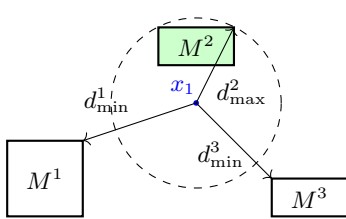
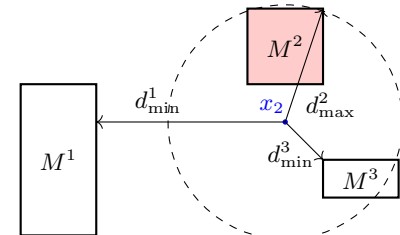

(a) Since $d^2_{\max} < \min\{d^1_{\min}, d^3_{\min}\} \implies b_{s,2} = 1$

(b) No forced assignment (L3.2 does not apply)

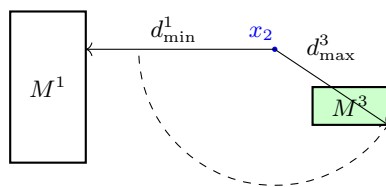
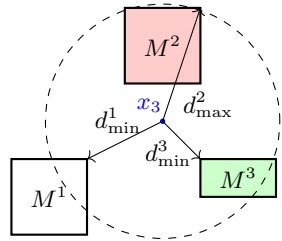

(c) Cannot-link: $b_{s,2} = 0 \Rightarrow$ L3.2 fixes $b_{s,3} = 1$

(d) Must-link: link propagation fixes $b_{s,3} = 1$

Figure 1: Illustration of sample-determination via link propagation for $K = 3$.

Figure 1 illustrates interaction variations of geometric checks and pairwise constraints for data points $x_1, x_2$ and $x_3$. In (a), the distance bounds immediately fix $x_1$ in $M^2$. In (b), the bounds overlap, distances are inconclusive and no assignment is made. In (c), the cannot–link $(x_1, x_2)$ rules out $M^2$ for $x_2$, after which the geometric test fixes $x_2$ in $M^3$. In (d), the must–link $(x_2, x_3)$ propagates that assignment to $x_3$. This sequence shows how geometry and ML/CL constraints jointly determine assignments prior to branching. If the ML/CL constraints forbid the move $(x_s \rightarrow k^+)$, the node becomes infeasible and is pruned.

### 3.2 EQUIVALENT UNCONSTRAINED CLUSTERING PROBLEM

Although the geometric sample-determination rules and link propagation of Section 3.1 eliminate most binary assignments, the remaining must-link constraints still couple samples and inflate the **branch–and–bound** (BB) complexity. To isolate this effect, consider the ML-only version of Problem (2). Different from the *unconstrained* MSSC problem, pairwise constraint clustering problem contains a family of equalities $b_{s,k} = b_{s',k}$ for $(s, s') \in \mathcal{T}_{ml}$. Nevertheless, we show that collapsing each must-link component into repeated pseudo-samples yields an unconstrained instance with identical global optimum.

Let a cluster $\mathcal{C} = \{x_1, x_2, \ldots, x_p\}$ and let $\mu$ denote an arbitrary centroid for $\mathcal{C}$. Without loss of generality, assume $x_1, \ldots, x_t \in \mathcal{C}_{ml} \subseteq \mathcal{C}$ form a single *must-link component* inside $\mathcal{C}$. Let $\mu_{ml}$ denote the centroid of $\mathcal{C}_{ml}$ and $tr(\Sigma^2_{ml})$ the trace of the covariance matrix of $\mathcal{C}_{ml}$. Thus, we have: $\mu_{ml} = \frac{1}{t} \sum_{i=1}^t x_i, \quad tr(\Sigma^2_{ml}) = \frac{1}{t-1} \sum_{i=1}^t ||x_i - \mu_{ml}||^2$.

**Lemma 3.3.** *Given 2 clusters,* $\mathcal{C} = \{x_1, x_2, \ldots, x_p\}$ *and* $\hat{\mathcal{C}} = \{x_{t+1}, x_{t+2}, \ldots, x_p, \underbrace{\mu_{ml}, \ldots, \mu_{ml}}_{t}\}$,

*let* $\mathsf{sse}_{\mathcal{C}}(\mu)$ *and* $\mathsf{sse}_{\hat{\mathcal{C}}}(\mu)$ *denote their respective within-cluster SSE computed with centroid* $\mu$. *Then, the following identity holds:*

$$\mathsf{sse}_{\mathcal{C}}(\mu) = \mathsf{sse}_{\hat{\mathcal{C}}}(\mu) + (t - 1)tr(\Sigma^2_{ml}). \tag{3}$$

Note that $(t-1)tr(\Sigma_{ml}^2)$ is an additive constant that does not affect optimization over centroids. Figure 2 illustrates this construction of pseudo-samples using a small example. Based on Lemma 3.3, we form the dataset:

$$\hat{X} = \bigcup_{k \in \mathcal{K}_{ml}} \{\underbrace{\mu_{ml,k}, \dots, \mu_{ml,k}}_{t_k}\} \cup (X \setminus \{x_s | (s, s') \in \mathcal{T}_{ml}, s' \in \mathcal{S}\}),$$

along with the corresponding unconstrained MSSC optimization problem:

$$\min_{\mu,d,b} \sum_{s \in \hat{\mathcal{S}}} d_{s,*} + \sum_{k \in \mathcal{K}_{ml}} (t_k - 1)tr(\Sigma_{ml,k}^2) \tag{4a}$$

$$\text{s.t. Constraints 2b, 2c, 1e, 1d} \tag{4b}$$

where $\mu_{ml,k}$ and $\Sigma_{ml,k}^2$ represent the mean (centroid) and covariance matrix of the must-link samples within cluster $k$, respectively. $\mathcal{K}_{ml}$ denotes the set of clusters containing must-link samples, and $\hat{\mathcal{S}}$ is the corresponding index set for the dataset $\hat{X}$.

**Theorem 3.4.** *If $\mu^*$ and $z(\mu^*)$ are the global optimal solution and cost of Problem* (4)*, then they are also the global optimum and cost of the ML–only problem* (2)*, and vice versa.*

**Mixed ML and CL constraints.** When both ML and CL constraints are present, collapse each ML component as above to obtain $\hat{X}$ with only CL constraints. Then apply Lemma 3.1 to eliminate all CL-violating assignments, yielding an unconstrained MSSC on the reduced dataset.

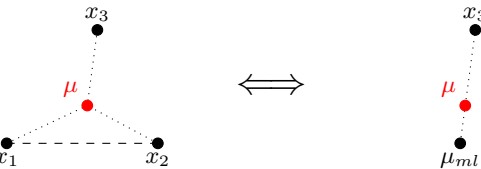

Figure 2: Left: three samples $x_1, x_2, x_3$ with must-link $(x_1, x_2)$ and centroid $\mu$. Right: collapse of $\{x_1, x_2\}$ into two pseudo-samples at $\mu_{ml} = \frac{1}{2}(x_1 + x_2)$ preserves the optimal centroid $\mu$.

### 3.3 UPPER BOUNDING STRATEGIES

Let $M_0 \subset \mathbb{R}^{mk}$ be the initial axis-aligned box for the full centroid vector $\mu = \{\mu_1, \dots, \mu_K\}$. The branch–and–bound (BB) algorithm requires a fast yet tight lower bound (LB) for each subproblem or node $M$ inside the solution space $M_0$. In this section two methods for computing upper bounds at each node of the BB scheme are presented. The first method handles CL constraints through a $k$–coloring interpretation. The second method derives a closed-form expression applicable when only ML constraints are present. When both constraint types coexist, ML constraints can be collapsed as described in Section 3.2, reformulating the problem into one involving only CL constraints and allowing the use of the $k$–coloring approach.

$K$–**Coloring Bound for CL Constraints.** Let $K$ denote the prescribed number of clusters and let $G_{cl} = (\hat{\mathcal{S}}, \mathcal{T}_{cl})$ be the CL graph. At node $M \subseteq M_0$ maintain a pool of $C$ centroid candidates $\{\mu^{(c)}\}_{c=1}^C \subset M$. For each candidate $c$ assign each sample $s \in \hat{\mathcal{S}}$ to its closest centroid, $k_s^{(c)} = \arg\min_{k \in \{1,\dots,K\}} \|x_s - \mu_k^{(c)}\|_2^2$, and let $\chi^{(c)}(s) = k_s^{(c)}$. The labeling $\chi^{(c)}$ is *proper* if $\chi^{(c)}(u) \neq \chi^{(c)}(v)$ for every $(u, v) \in \mathcal{T}_{cl}$. Define

$$z_{ub}^{(c)} = \begin{cases} \sum_{s \in \hat{\mathcal{S}}} \|x_s - \mu_{\chi^{(c)}(s)}^{(c)}\|_2^2, & \text{if } \chi^{(c)} \text{ is proper,} \\ +\infty, & \text{otherwise.} \end{cases}$$

The node upper bound is $\alpha(M) = \min_{1 \leq c \leq C} z_{ub}^{(c)}$. If $\chi^{(c)}$ is proper, then $z_{ub}^{(c)}$ satisfies all CL constraints and $z(M) \leq \alpha(M)$.

**Closed–Form Bound for ML Constraints.** With only must–link (ML) constraints, fix any feasible centroid set $\hat\mu \in M$ and compute $\alpha(M) = \sum_{s \in \hat{\mathcal{S}}} Q_s(\hat\mu)$, where each $Q_s(\hat\mu)$ admits a closed form. The expression yields an admissible bound because every $\hat\mu \in M$ respects the ML constraints, implying $z(M) \leq \alpha(M)$. An initial bound is produced at the root via a heuristic ($k$-means). Bounds at descendant nodes are updated with candidates extracted from the relaxations in Section 3.4. The BB algorithm terminates with the optimal objective value $\alpha + \sum_{k \in \mathcal{K}_{\mathrm{ml}}} q_k \sigma^2_{\mathrm{ml},k}$.

### 3.4 Lower Bounding Strategy with Grouping-based Lagrangian Decomposition

The branch–and–bound (BB) algorithm requires a fast yet tight lower bound (LB) for each sub-problem or node $M$ inside the solution space $M_0$. An effective strategy to achieve tighter lower bounds is through Lagrangian decomposition (LD), in which the corresponding non-anticipativity constraints Cao & Zavala (2019) are dualized with fixed Lagrange multipliers $\lambda$ and added to the objective function (Karuppiah & Grossmann, 2008). However, to reduce problem size and improve relaxation quality, instead of associating each sample with a separate subproblem (as mentioned in Karuppiah & Grossmann (2008)), we partition the sample set $\hat{\mathcal{S}}$ into $G$ disjoint groups $\hat{\mathcal{S}}_1, \ldots, \hat{\mathcal{S}}_G$ with index set $\mathcal{G} = 1, \ldots, G$, such that $\bigcup_g \hat{\mathcal{S}}_g = \hat{\mathcal{S}}$ and $\hat{\mathcal{S}}_i \cap \hat{\mathcal{S}}_g = \emptyset$ for $i \neq g$. Instead of replicating center variables per sample, we assign one per group and enforce consistency through:

$$\min_{\mu_g \in M} \sum_{g \in \mathcal{G}} Q_g(\mu_g), \quad Q_g(\mu_g) := \sum_{s \in \hat{\mathcal{S}}_g} Q_s(\mu_g) \quad \text{s.t.} \quad \mu_g = \mu_{g+1}, \quad \forall g \in 1, \ldots, G-1 \quad (5a)$$

Dualizing the coupling constraints with multipliers $\lambda$ yields a tighter lower bound via:

$$\beta^{SG+LD}(M) := \max_\lambda \beta^{SG+LD}(M, \lambda) \quad (6)$$

This grouped formulation preserves intra-group non-anticipativity while relaxing inter-group consistency, yielding $\beta^{LD}(M) \leq \beta^{SG+LD}(M) \leq z(M)$. While solving (6) requires iterative MISOCPs, it significantly strengthens the bound. The grouping is fixed at the BB root for efficiency.

### 3.5 Branch-and-Bound Clustering Scheme

We adopt the framework of the reduced-space branch-and-bound scheme from (Cao & Zavala, 2019) and tailor the algorithm for the pairwise constrained clustering task. Algorithm 1 depicts the details of the algorithm, where $\beta$ and $\alpha$ represent the function of lower and upper bound, respectively. With the lower and upper bounding strategy provided in the following subsections.

**Theorem 3.5.** *Given an exhaustive subdivision on $\mu$, Algorithm 1 converges in the sense that*

$$\lim_{i \to \infty} \alpha_i = \lim_{i \to \infty} \beta_i = z. \quad (7)$$

The proof is shown in Appendix C.

## 4 Computational Experiments

We implemented our algorithm, **Sample-Driven Constrained Group-Based Branch-and-Bound (SDC-GBB)**, in `Julia 1.10.3` and evaluated its performance on synthetic and real-world datasets using a high-performance cluster comprising nodes with 128 AMD Epyc 7702 CPUs (2.0GHz) and 1TB of RAM. Computational experiments were conducted under both serial and parallel configurations, comparing SDC-GBB against the branch-and-bound (BB) algorithm in CPLEX 22.1.1 (Cplex, 2022), the exact method PC-SOS-SDP (Piccialli et al., 2022a), and the best heuristic out of the following algorithms: COP-$k$-means (Wagstaff et al., 2001), encode-kmeans-post (Nghiem et al., 2020), BLPKM-CC (Baumann, 2020), and Sensitivity Sampling coreset algorithm (Feldman & Langberg, 2011). All heuristic algorithms were run with 100 restarts (Wagstaff et al., 2001) in 4 hours, and the full results are reported in Appendix E. For parallel applications, subproblems were distributed across multiple CPU cores with group sizes limited to $\min(162/d - k, 10 \times k)$ during the lower bound decomposition process. Performance was assessed using Upper Bound (UB), relative optimality gap, and the number of BB nodes resolved. UB represents the best feasible solution found,

---

**Algorithm 1** Branch-and-Bound Clustering with Geometric Sample Determination

---

**Inputs:** $X = \{x_s\}_{s \in S} \subset \mathbb{R}^d$, $K$, $\mathcal{T}_{ml}$, $\mathcal{T}_{cl}$
**Initialization**
Initialize $i = 0$, $\mathbb{M} \leftarrow \{M_0\}$, tolerance $\epsilon > 0$
Compute upper bound $\alpha_i = \alpha(M_0)$, lower bound $\beta_i = \beta(M_0)$;
**Geometric Sample Determination**
Compute $d_{\min}(x_s, M_{0,k})$, $d_{\max}(x_s, M_{0,k})$;
**if** $d_{\min}(x_s, M_{0,k}) > \alpha_i$ **then** $b_{s,k} \leftarrow 0$ (update $K_s$);
**if** $\{\exists k^+$ with $d_{\max}(x_s, M_{0,k^+}) < \min_{k \neq k^+} d_{\min}(x_s, M_{0,k})\}$ **then** $b_{s,k^+} \leftarrow 1$;
Propagate fixes via $\mathcal{T}_{ml}$, $\mathcal{T}_{cl}$; update $\{K_s\}$;
**repeat**
    **Node Selection**
    Select a set $M \in \mathbb{M}$ satisfying $\beta(M) = \beta_i$;
    $\mathbb{M} \leftarrow \mathbb{M} \setminus \{M\}$;
    $i \leftarrow i + 1$;
    **Branching**
    Partition $M$ into subsets $M_1$ and $M_2$ with $relint(M_1) \cap relint(M_2) = \emptyset$;
    Add each subset to $\mathbb{M}$ to create separated child nodes;
    **Bounding**
    Compute $\alpha(M_1)$, $\beta(M_1)$, $\alpha(M_2)$, $\beta(M_2)$;
    $\beta_i \leftarrow \min\{\beta(M') \mid M' \in \mathbb{M}\}$;
    $\alpha_i \leftarrow \min\{\alpha_{i-1}, \alpha(M_1), \alpha(M_2)\}$;
    Remove all $M'$ from $\mathbb{M}$ if $\beta(M') \geq \alpha_i$;
    If $|\beta_i - \alpha_i| \leq \epsilon$, STOP;
**until** $\mathbb{M} = \emptyset$
**Output** $\hat{\mu}, \hat{b}$ and $z^\star = \alpha_i + \sum_{k \in \mathcal{K}_{ml}} (t_k - 1) \, tr(\Sigma_{ml,k}^2)$

---

while the relative optimality gap is calculated as $\frac{\alpha_l - \beta_l}{\min(\alpha_l, \beta_l)} \times 100\%$, where $\alpha_l$ and $\beta_l$ denote the best lower and upper bounds, respectively. The number of resolved BB nodes indicates the total BB iterations performed. Unlike heuristic methods, deterministic global optimization methods provide an optimality gap, enabling quantitative assessment of solution quality.

We evaluate the selected algorithms on 8 real-world datasets taken or sampled from the UCI Machine Learning Repository (Dua & Graff, 2017), Hemicellulose (Wang et al., 2022), PR2392 (Padberg & Rinaldi, 1991), and 7 synthetic datasets generated with 2 features, 3 Gaussian clusters and a fixed random seed (seed = 1). Datasets are categorized as small ($n \leq 1,000$), medium ($n \leq 10,000$), large ($n \leq 100,000$), and huge ($n \geq 100,000$), where $n$ is the number of samples [1]. We follow the pairwise constraint generation practice with the same classic random-pair sampling pipeline as (Piccialli et al., 2022a; Aloise et al., 2009; Babaki et al., 2014; Guns et al., 2016). Across three separate experiments, namely must-link only (ML-only), cannot-link only (CL-only), and both must-link and cannot-link (ML+CL), each dataset has $\frac{n}{4}$ samples bounded by ML constraints, $\frac{n}{4}$ samples bounded by CL constraints, and combining $\frac{n}{4}$ ML with $\frac{n}{4}$ CL constraints respectively. when the optimality gap dropped below 0.1 %, when runtime reached 4 hours for datasets with $n \leq 10,000$ or 12 h for those with $n > 10,000$, or when 5 million nodes had been explored.

## 4.1 NUMERICAL RESULTS

Our work focuses explicitly on solution quality in terms of the MSSC cost, providing an optimality guarantee. With this, SDC-GBB matches the performance of commercial solvers and the state-of-the-art algorithm PC-SOS-SDP (Piccialli et al., 2022a) through *super-point aggregation, targeted decomposition, and tight bounding via geometric partitioning* under ML constraints, as well as *geometric sample determination rules* that prune infeasible assignments under CL constraints. SDC-GBB successfully handles instances exceeding two hundred thousand samples across all constraint cases and further scales to ML-only instances with more than 1.5 million samples.

---

[1]Tables 2-3 use a subset of datasets from Table 1 ($n \leq 210,000$). Table 1 includes 15 datasets of up to 1.5M samples that are only tractable under ML constraints due to the scalability limitations discussed in Section 6.

**Small and medium-sized datasets** For small datasets, SDC-GBB matches the state-of-the-art PC-SOS-SDP algorithm, outperforming CPLEX and heuristic methods across all constraint settings, as shown in Tables 1, 2, 3. On real-world benchmarks, SDC-GBB and PC-SOS-SDP achieve global optimality with gaps $\leq 0.1\%$ on Iris ($n = 150$) (Fisher, 1936) and Seeds ($n = 210$) (Charytanowicz et al., 2010). The heuristic method also closely approximates the global optima, whereas CPLEX yields significantly larger gaps of around 10%–69% for ML, 73%–86% for CL, and 37%–75% for combined constraints. In experiments with medium-sized datasets, SDC-GBB consistently outperforms all other algorithms, achieving optimality gaps $\leq 0.1\%$ in nearly all cases, with exceptions in PR2392 (CL-only) and RDS_CNT (CL-only), where gaps slightly increase to 2.68% and 1.23%, respectively. The best heuristic method consistently performs slightly worse than SDC-GBB but remains competitive, providing relatively small gaps across datasets. Conversely, CPLEX returns gaps close to 100% across all medium-sized datasets, and PC-SOS-SDP either generates higher gaps than SDC-GBB or fails to converge for datasets with $\geq 2,000$ samples.

Table 1: Computational performance with must-link (SDC-GBB, $k = 3$).

| DATASET | METHOD | UB | NODES | GAP(%) | DATASET | METHOD | UB | NODES | GAP(%) |
|---|---|---|---|---|---|---|---|---|---|
| REAL-WORLD DATASETS | | | | | | | | | |
| IRIS[2] | HEURISTIC | 83.82 | – | – | HTRU2 | HEURISTIC | $1.407 \times 10^8$ | – | – |
| N = 150 | CPLEX | 84.07 | 12987400 | 10.55% | N = 17,898 | CPLEX | NO FEASIBLE SOLUTION FOUND | | |
| D = 4 | PC-SOS-SDP | **83.63** | 1 | $\leq$ **0.1%** | D = 8 | PC-SOS-SDP | NO SOLUTION FOUND | | |
| | PARALLEL | **83.63** | 10 | $\leq$ **0.1%** | | PARALLEL | $\mathbf{1.022 \times 10^8}$ | 67 | $\leq$ **0.1%** |
| SEED[2] | HEURISTIC | 620.78 | – | – | SPNET3D_5[3] | HEURISTIC | $6.627 \times 10^6$ | – | – |
| N = 210 | CPLEX | 755.91 | 5814200 | 69.48% | N = 50,000 | CPLEX | NO FEASIBLE SOLUTION FOUND | | |
| D=7 | PC-SOS-SDP | **620.23** | 1 | $\leq$ **0.1%** | D = 3 | PC-SOS-SDP | NO SOLUTION FOUND | | |
| | PARALLEL | **620.23** | 7 | $\leq$ **0.1%** | | PARALLEL | $\mathbf{6.609 \times 10^6}$ | 61 | 1.51% |
| HEMI[2] | HEURISTIC | $1.602 \times 10^7$ | – | – | SKIN_8 | HEURISTIC | $4.533 \times 10^8$ | – | – |
| N = 1,955 | CPLEX | $3.204 \times 10^7$ | 90400 | 96.26% | N = 80,000 | CPLEX | NO FEASIBLE SOLUTION FOUND | | |
| D = 7 | PC-SOS-SDP | $1.601 \times 10^7$ | 1 | 2.07% | D = 3 | PC-SOS-SDP | NO SOLUTION FOUND | | |
| | PARALLEL | $\mathbf{1.601 \times 10^7}$ | 8 | $\leq$ **0.1%** | | PARALLEL | $\mathbf{4.138 \times 10^8}$ | 12 | 0.62% |
| PR2392[2] | HEURISTIC | $3.210 \times 10^{10}$ | – | – | URBANGB | HEURISTIC | $1.643 \times 10^9$ | – | – |
| N = 2,392 | CPLEX | $3.816 \times 10^{10}$ | 281000 | 98.76% | N = 360,177 | CPLEX | NO FEASIBLE SOLUTION FOUND | | |
| D = 2 | PC-SOS-SDP | NO SOLUTION FOUND | | | D = 2 | PC-SOS-SDP | NO SOLUTION FOUND | | |
| | PARALLEL | $\mathbf{3.209 \times 10^{10}}$ | 22 | $\leq$ **0.1%** | | PARALLEL | $\mathbf{4.135 \times 10^5}$ | 2 | **12.97%** |
| RDS_CNT[2] | HEURISTIC | $6.122 \times 10^7$ | – | – | SPNET3D | HEURISTIC | $5.848 \times 10^7$ | – | – |
| N = 10,000 | CPLEX | $1.154 \times 10^8$ | 12600 | 100.00% | N = 434,874 | CPLEX | NO FEASIBLE SOLUTION FOUND | | |
| D = 3 | PC-SOS-SDP | NO SOLUTION FOUND | | | D = 3 | PC-SOS-SDP | NO SOLUTION FOUND | | |
| | PARALLEL | $\mathbf{6.078 \times 10^7}$ | 32 | $\leq$ **0.1%** | | PARALLEL | $\mathbf{5.797 \times 10^7}$ | 2 | 7.43% |
| SYNTHETIC DATASETS (D = 2) | | | | | | | | | |
| SYN-210000 | HEURISTIC | $2.163 \times 10^6$ | – | – | SYN-1050000 | HEURISTIC | $1.050 \times 10^7$ | – | – |
| N = 210,000 | CPLEX | NO FEASIBLE SOLUTION FOUND | | | N = 1,050,000 | CPLEX | NO FEASIBLE SOLUTION FOUND | | |
| | PC-SOS-SDP | NO SOLUTION FOUND | | | | PC-SOS-SDP | NO SOLUTION FOUND | | |
| | PARALLEL | $\mathbf{2.163 \times 10^6}$ | 26 | $\leq$ **0.1%** | | PARALLEL | $\mathbf{1.050 \times 10^7}$ | 1 | 2.45% |
| SYN-420000 | HEURISTIC | $6.010 \times 10^6$ | – | – | SYN-1500000 | HEURISTIC | $1.725 \times 10^7$ | – | – |
| N = 420,000 | CPLEX | NO FEASIBLE SOLUTION FOUND | | | N = 1,500,000 | CPLEX | NO FEASIBLE SOLUTION FOUND | | |
| | PC-SOS-SDP | NO SOLUTION FOUND | | | | PC-SOS-SDP | NO SOLUTION FOUND | | |
| | PARALLEL | $\mathbf{6.010 \times 10^6}$ | 18 | 0.61% | | PARALLEL | $\mathbf{1.725 \times 10^7}$ | 1 | 2.66% |

Table 2: Computational performance with cannot-link (SDC-GBB, $k = 3$).

| DATASET | METHOD | UB | NODES | GAP(%) | DATASET | METHOD | UB | NODES | GAP(%) |
|---|---|---|---|---|---|---|---|---|---|
| REAL-WORLD DATASETS | | | | | | | | | |
| IRIS[2] | HEURISTIC | 80.31 | – | – | RDS_CNT[2] | HEURISTIC | $2.897 \times 10^7$ | – | – |
| N = 150 | CPLEX | 119.03 | 8259900 | 73.59% | N = 10,000 | CPLEX | $5.198 \times 10^7$ | 11559 | 100.00% |
| D = 4 | PC-SOS-SDP | **80.21** | 1 | $\leq$ **0.1%** | D = 3 | PC-SOS-SDP | NO SOLUTION FOUND | | |
| | PARALLEL | **80.21** | 17 | $\leq$ **0.1%** | | PARALLEL | $\mathbf{2.861 \times 10^7}$ | 49 | 1.23% |
| SEED[2] | HEURISTIC | 603.04 | – | – | HTRU2[3] | HEURISTIC | $1.740 \times 10^8$ | – | – |
| N = 210 | CPLEX | 771.54 | 5244683 | 86.67% | N = 17,898 | CPLEX | NO FEASIBLE SOLUTION FOUND | | |
| D = 7 | PC-SOS-SDP | **601.96** | 15 | $\leq$ **0.1%** | D = 8 | PC-SOS-SDP | NO SOLUTION FOUND | | |
| | PARALLEL | **601.96** | 18 | $\leq$ **0.1%** | | PARALLEL | $\mathbf{9.225 \times 10^7}$ | 15 | $\leq$ **0.1%** |
| HEMI[2] | HEURISTIC | $1.401 \times 10^7$ | – | – | SPNET3D_5 | HEURISTIC | $3.938 \times 10^6$ | – | – |
| N = 1,955 | CPLEX | $2.667 \times 10^7$ | 65002 | 100.00% | N = 50,000 | CPLEX | NO FEASIBLE SOLUTION FOUND | | |
| D = 7 | PC-SOS-SDP | $1.328 \times 10^7$ | i[4] | 17.70% | D = 3 | PC-SOS-SDP | NO SOLUTION FOUND | | |
| | PARALLEL | $\mathbf{1.328 \times 10^7}$ | 5 | $\leq$ **0.1%** | | PARALLEL | $\mathbf{3.831 \times 10^6}$ | 34 | $\leq$ **0.1%** |
| PR2392[2] | HEURISTIC | $2.566 \times 10^{10}$ | – | – | SKIN_8 | HEURISTIC | $6.367 \times 10^8$ | – | – |
| N = 2,392 | CPLEX | $3.266 \times 10^{10}$ | 256964 | 99.96% | N = 80,000 | CPLEX | NO FEASIBLE SOLUTION FOUND | | |
| D = 2 | PC-SOS-SDP | NO SOLUTION FOUND | | | D = 3 | PC-SOS-SDP | NO SOLUTION FOUND | | |
| | PARALLEL | $\mathbf{2.512 \times 10^{10}}$ | 97 | 2.68% | | PARALLEL | $\mathbf{3.016 \times 10^8}$ | 8 | 1.32% |
| SYNTHETIC DATASETS (D = 2) | | | | | | | | | |
| SYN-12000 | HEURISTIC | $9.503 \times 10^4$ | – | – | SYN-42000 | HEURISTIC | $5.116 \times 10^5$ | – | – |
| N = 12,000 | CPLEX | NO FEASIBLE SOLUTION FOUND | | | N = 42,000 | CPLEX | NO FEASIBLE SOLUTION FOUND | | |
| | PC-SOS-SDP | NO SOLUTION FOUND | | | | PC-SOS-SDP | NO SOLUTION FOUND | | |
| | PARALLEL | $\mathbf{9.053 \times 10^4}$ | 89 | 0.75% | | PARALLEL | $\mathbf{5.116 \times 10^5}$ | 22 | $\leq$ **0.1%** |
| SYN-21000 | HEURISTIC | $1.817 \times 10^5$ | – | – | SYN-210000 | HEURISTIC | $2.161 \times 10^6$ | – | – |
| N = 21,000 | CPLEX | NO FEASIBLE SOLUTION FOUND | | | N = 210,000 | CPLEX | NO FEASIBLE SOLUTION FOUND | | |
| | PC-SOS-SDP | NO SOLUTION FOUND | | | | PC-SOS-SDP | NO SOLUTION FOUND | | |
| | PARALLEL | $\mathbf{1.817 \times 10^5}$ | 27 | 0.32% | | PARALLEL | $\mathbf{2.161 \times 10^6}$ | 1 | 2.38% |

**Large and huge datasets** Only SDC-GBB and heuristic algorithms can handle $n > 10,000$ datasets, with SDC-GBB getting better UB and reaching global optimality or maintaining stable gaps below 3% in all datasets and experiments other than URBANGB and SPNET3D, which receive 5% - 13% gaps. Although the gaps we achieved with these instances are not optimal, they can be further optimized with increased parallelization, and it is worth noting that no other global optimization method can find solutions at this scale. Meanwhile, both PC-SOS-SDP and CPLEX cannot find any feasible solution for these datasets given the 12-hour time limit. This shows that SDC-GBB can scale up to 1,500 times for ML-only constraints, and 200 times for CL-only and ML+CL constraints when compared with state-of-the-art algorithms.

Table 3: Computational performance with both must-link and cannot-link (SDC-GBB, $k = 3$).

| | | REAL-WORLD DATASETS | | | | | | | |
|---|---|---|---|---|---|---|---|---|---|
| DATASET | METHOD | UB | NODES | GAP(%) | DATASET | METHOD | UB | NODES | GAP(%) |
| IRIS[2] | HEURISTIC | 86.85 | – | – | RDS_CNT[2] | HEURISTIC | $7.579 \times 10^7$ | – | – |
| N = 150 | CPLEX | 93.75 | 9166269 | 37.47% | N = 10,000 | CPLEX | $1.493 \times 10^8$ | 4100 | 100.00% |
| D = 4 | PC-SOS-SDP | **86.76** | 3 | $\leq$ **0.1%** | D = 3 | PC-SOS-SDP | NO SOLUTION FOUND | | |
| | PARALLEL | **86.76** | 17 | $\leq$ **0.1%** | | PARALLEL | $\mathbf{7.437 \times 10^7}$ | 32 | $\leq$ **0.1%** |
| SEED[2] | HEURISTIC | 597.14 | – | – | HTRU2[3] | HEURISTIC | $1.859 \times 10^8$ | – | – |
| N = 210 | CPLEX | 760.73 | 6044677 | 75.32% | N = 17,898 | CPLEX | NO FEASIBLE SOLUTION FOUND | | |
| D = 7 | PC-SOS-SDP | **596.61** | 5 | $\leq$ **0.1%** | D = 8 | PC-SOS-SDP | NO SOLUTION FOUND | | |
| | PARALLEL | **596.61** | 9 | $\leq$ **0.1%** | | PARALLEL | $\mathbf{1.141 \times 10^8}$ | 79 | $\leq$ **0.1%** |
| HEMI[2] | HEURISTIC | $1.566 \times 10^7$ | – | – | SPNET3D_5[3] | HEURISTIC | $8.171 \times 10^6$ | – | – |
| N = 1,955 | CPLEX | $3.519 \times 10^7$ | 39600 | 100.00% | N = 50,000 | CPLEX | NO FEASIBLE SOLUTION FOUND | | |
| D = 7 | PC-SOS-SDP | NO SOLUTION FOUND | | | D = 3 | PC-SOS-SDP | NO SOLUTION FOUND | | |
| | PARALLEL | $\mathbf{1.533 \times 10^7}$ | 115 | $\leq$ **0.1%** | | PARALLEL | $\mathbf{7.925 \times 10^6}$ | 45 | **5.01%** |
| PR2392[2] | HEURISTIC | $2.922 \times 10^{10}$ | – | – | SKIN_8 | HEURISTIC | $7.579 \times 10^8$ | – | – |
| N = 2,392 | CPLEX | $3.240 \times 10^{10}$ | 239465 | 99.50% | N = 80,000 | CPLEX | NO FEASIBLE SOLUTION FOUND | | |
| D = 2 | PC-SOS-SDP | NO SOLUTION FOUND | | | D = 3 | PC-SOS-SDP | NO SOLUTION FOUND | | |
| | PARALLEL | $\mathbf{2.916 \times 10^{10}}$ | 35 | $\leq$ **0.1%** | | PARALLEL | $\mathbf{4.258 \times 10^8}$ | 11 | **4.86%** |
| | | SYNTHETIC DATASETS (D = 2) | | | | | | | |
| SYN-12000 | HEURISTIC | $9.520 \times 10^4$ | – | – | SYN-42000 | HEURISTIC | $5.133 \times 10^5$ | – | – |
| N = 12,000 | CPLEX | NO FEASIBLE SOLUTION FOUND | | | N = 42,000 | CPLEX | NO FEASIBLE SOLUTION FOUND | | |
| | PC-SOS-SDP | NO SOLUTION FOUND | | | | PC-SOS-SDP | NO SOLUTION FOUND | | |
| | PARALLEL | $\mathbf{9.520 \times 10^4}$ | 25 | $\leq$ **0.1%** | | PARALLEL | $\mathbf{5.133 \times 10^5}$ | 31 | **1.37%** |
| SYN-21000 | HEURISTIC | $1.818 \times 10^5$ | – | – | SYN-210000 | HEURISTIC | $2.165 \times 10^6$ | – | – |
| N = 21,000 | CPLEX | NO FEASIBLE SOLUTION FOUND | | | N = 210,000 | CPLEX | NO FEASIBLE SOLUTION FOUND | | |
| | PC-SOS-SDP | NO SOLUTION FOUND | | | | PC-SOS-SDP | NO SOLUTION FOUND | | |
| | PARALLEL | $\mathbf{1.818 \times 10^5}$ | 37 | 0.18% | | PARALLEL | $\mathbf{2.164 \times 10^6}$ | 28 | **0.64%** |

[2] LESS THAN 4 HOURS.
[3] LESS THAN 8 HOURS.
[4] SOLVED AT THE ROOT NODE.

## 5 CONCLUSION

In this paper, we presented Sample-Driven Constrained Group-Based Branch-and-Bound (SDC-GBB), a deterministic global optimization algorithm for pairwise-constrained MSSC. We prove convergence to a globally $\varepsilon$-optimal solution and demonstrate scalability to datasets exceeding 200,000 samples in all constraint settings, which is **over 200 times larger** than the 800-sample benchmark of (Piccialli et al., 2022a), and further extend to **over 1,500 times larger** with ML-only instances having more than one million samples while maintaining optimality gaps below 3%. When empirically evaluated on real-world benchmarks of various domains, SDC-GBB consistently achieves low optimality gaps across diverse constraint settings.

## 6 LIMITATIONS AND FUTURE DIRECTIONS

Similar to prior work, our algorithm struggles to scale to one million samples with CL constraints due to the NP-hard nature of this constraint type. Future work may consider tightening the grouped-sample Lagrangian lower bound (Section 3.4) by dualizing the CL graph with relaxing binary indicators to [0,1] and penalty multipliers $z_{ij}$ updated via subgradient methods. This formulation would produce a more compact MISOCP with fewer active binary variables thanks to the relaxation of $z_{ij}$ and stronger continuous bounds, reducing the number of branches in dense CL graphs, as shown by successful Lagrangian-penalty approaches in semi-supervised clustering and global MISOCP strategies. An alternative is to incorporate clique inequalities or separation cuts for the CL graph, following MIP methodologies that substantially improve bounds. However, including these Lagrangian terms at each node would increase the solve time of the relaxation, so empirically evaluating the trade-off between bound improvement and per-node cost will be key.

**Ethics Statement**  Our study uses the SKIN (Skin Segmentation) dataset from the UCI Machine Learning Repository. The dataset contains de-identified RGB pixel triplets (B, G, R) sampled from facial images and is derived from two collections: the PAL Face Database and the DARPA-sponsored Color FERET images. All human participants provided consent for the use of these collections for research purposes. In line with data-privacy best practices, the UCI release exposes only anonymized pixel triplets and does not include raw images, which reduces the risk of re-identification in downstream work. Concurrently, our framework optimizes only the Minimum Sum-of-Squares Criteria (MSSC) objective, and as is well documented, MSSC/k-means–style clustering can reproduce and even amplify existing biases in the data, particularly at scale. In high-stakes or sensitive domains, deployments that do not account for these effects can lead to disparate treatment or outcomes across demographic groups. We recommend that practitioners perform fairness audits before deployment, and consider mitigation techniques such as fair clustering variants and post-processing adjustments if such disparities arise. The absence of fairness-aware safeguards in the current implementation is a limitation of this work, and integrating such constraints or corrections is left for future extensions.

**Reproducibility Statement**  In this section, we outline necessary details for reproducing all experiments described in the paper. We provide comprehensive hardware and software configuration for SDC-GBB as well as pairwise constraint generation, seeding protocol, runtime limit and data setup, including real and synthetic dataset generation in Section 4. The implementation of the SDC-GBB algorithm is thoroughly described through Algorithm 1. Lastly, we document evaluation in Section 4 along with the results of all baselines used for comparison in Sections D and E. To further support the reproducibility of our results, we will release our experiment code upon acceptance, enabling other researchers to replicate and expand on our work.

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

## A    TWO-STAGE PROGRAMS REFORMULATION

Once the must-link set $\mathcal{T}_{ml}$ has been collapsed into pseudo-samples (Sec. 3.2), the additive constant $(t-1)\operatorname{tr}(\Sigma_{ml}^2)$ can be pre-computed. Hence minimising the objective in (4a) is equivalent to minimising $\sum_{s\in\hat{\mathcal{S}}} d_{s,*}$. The optimal solution of (4) is obtained from the two-stage program:

$$z \;=\; \min_{\mu\in M_0} \sum_{s\in\hat{\mathcal{S}}} Q_s(\mu), \tag{8}$$

where $\mu$ are the *first-stage* variables and $Q_s(\mu)$ is the optimal value of the *second-stage* problem defined below. After (i) collapsing must-link components and (ii) applying the geometric Lemmas 1–2 together with cannot-link propagation, each sample $s$ may still be assigned only to a *viable* subset $\mathcal{K}_s \subseteq \mathcal{K}$. The reduced dataset $\hat{\mathcal{S}}$ and the family $\{\mathcal{K}_s\}_{s\in\hat{\mathcal{S}}}$ are fixed once at the root node.

$$Q_s(\mu) = \min_{d_s, b_s}\; d_{s,*}$$
$$\text{s.t.  Constraints 2b, 2c, 1e, 1d.} \tag{9}$$

Here $d_s = [d_{s,k}]_{k\in\mathcal{K}_s}$ and $b_s = [b_{s,k}]_{k\in\mathcal{K}_s}$ are the *second-stage* variables. The closed set $M_0 = \{\mu \mid \mu^l \le \mu \le \mu^u\}$ bounds every centre with $\mu_{k,i}^l = \min_s X_{s,i}$ and $\mu_{k,i}^u = \max_s X_{s,i}$ for all $k \in \mathcal{K}$ and $i = 1,\dots,m$. For convenience we choose a *single* Big-$M$ constant

$$N = \max_{s,k} \sum_{i=1}^m \max\big\{|x_{s,i} - \mu_{k,i}^l|^2, |x_{s,i} - \mu_{k,i}^u|^2\big\},$$

which leaves all bounds valid and simplifies the notation. The bounds $\mu^l, \mu^u$ are computed once at the root node and inherited unchanged by every BB subproblem. We denote by $\operatorname{relint}(\mathcal{M})$ and $\delta(\mathcal{M})$ the relative interior and the diameter of a set. Throughout this paper, the diameter of the box set $M_0$ is $\delta(M_0) = ||\mu^u - \mu^l||_\infty$.

It can be shown that the closed-form solution to the second-stage problem is

$$Q_s(\mu) = \min_{k\in\mathcal{K}_s} \|x_s - \mu_k\|_2^2.$$

Since $Q_s(\mu)$ is the minimum of a finite number of continuous functions, $Q_s$ is continuous. Because of the compactness of $M_0$ and continuity of $Q_s(\mu)$, the clustering Problem 8 can attain its minimum according to the generalized Weierstrass theorem.

When the BB algorithm explores a box $M \subseteq M_0$, it solves the *primal node problem*

$$z(M) = \min_{\mu\in M} \sum_{s\in\hat{\mathcal{S}}} Q_s(\mu). \tag{10}$$

Replicating centres for each sample and enforcing non-anticipativity (11b) yields the lifted form

$$\min_{\mu_s\in M} \quad \sum_{s\in\hat{\mathcal{S}}} Q_s(\mu_s) \tag{11a}$$

$$\text{s.t. } \mu_s = \mu_{s+1}, \qquad s = 1,\dots,|\hat{\mathcal{S}}|-1. \tag{11b}$$

Problems (10) and (11) are equivalent, and retain all cannot-link information through the viable-cluster sets $\{\mathcal{K}_s\}_{s\in\hat{\mathcal{S}}}$.

# B PROOF OF THEOREMS

## B.1 PROOF OF LEMMA 3.3

*Proof.*

$$\text{sse}_{\mathcal{C}}(\mu) = \sum_{i=1}^{p} ||x_i - \mu||^2 \tag{12}$$

$$= \sum_{i=1}^{t} ||x_i - \mu||^2 + \sum_{i=t+1}^{p} ||x_i - \mu||^2 \tag{13}$$

$$= \sum_{i=1}^{t} ||x_i||^2 - 2\mu^T \sum_{i=1}^{t} x_i + t||\mu||^2 + \sum_{i=t+1}^{p} ||x_i - \mu||^2 \tag{14}$$

Here $\sum_{i=1}^{t} ||x_i||^2$ can be rewritten as follow:

$$\sum_{i=1}^{t} ||x_i||^2 = \sum_{i=1}^{t} ||x_i - \mu_{ml} + \mu_{ml}||^2 \tag{15}$$

$$= \sum_{i=1}^{t} ||x_i - \mu_{ml}||^2 - 2\mu_{ml}^T \sum_{i=1}^{t} (x_i - \mu_{ml}) + t||\mu_{ml}||^2 \tag{16}$$

$$= \sum_{i=1}^{t} ||x_i - \mu_{ml}||^2 + t||\mu_{ml}||^2 \tag{17}$$

$$= (t-1)tr(\Sigma^2) + t||\mu_{ml}||^2 \tag{18}$$

Thus, we have:

$$\text{sse}_{\mathcal{C}}(\mu) = \sum_{i=1}^{t} ||x_i||^2 - 2\mu^T \sum_{i=1}^{t} x_i + t||\mu||^2 + \sum_{i=t+1}^{p} ||x_i - \mu||^2 \tag{19}$$

$$= (t-1)tr(\Sigma_{ml}^2) + t||\mu_{ml}||^2 - 2t\mu^T\mu_{ml} + t||\mu||^2 + \sum_{i=t+1}^{p} ||x_i - \mu||^2 \tag{20}$$

$$= (t-1)tr(\Sigma_{ml}^2) + t||\mu_{ml} - \mu||^2 + \sum_{i=t+1}^{p} ||x_i - \mu||^2 \tag{21}$$

$$= (t-1)tr(\Sigma_{ml}^2) + \text{sse}_{\hat{\mathcal{C}}}(\mu) \tag{22}$$

$\square$

## B.2 PROOF OF THEOREM 3.4

*Proof.* ($\Rightarrow$) Let $\mu^*$ denote a globally optimal solution of Problem (4) as a result of ML collapse. Each pseudo-sample then corresponds to exactly one must-link component of the original dataset. In constructing Problem (4), assign every genuine sample in that component to the cluster of its associated pseudo-sample; any sample not belonging to a must-link component retains the label it receives in the unconstrained solution. This enforces $b_{i,k} = b_{i',k}$ for all $(i, i') \in \mathcal{T}_{ml}$, so the pair $(\mu^*, b^*)$ is feasible for Problem (2). At the same time, the resulting objective matches that of Problem (4), since the additive term $\sum_{k \in \mathcal{K}_{ml}} (t_k - 1) \text{tr}(\Sigma_{ml,k}^2)$ precisely reinstates the variance eliminated when collapsing each must-link component. Hence, the optimal solution of Problem (4) is optimal for Problem (2).

($\Leftarrow$) Let $(\mu^*, b^*)$ be a global optimum for Problem (2) on the original instance. Collapse ML constraints based on Lemma 2 to form the pseudo-sample instance $\hat{\mu}$. Then $\hat{\mu}$ is feasible for the unconstrained Problem (4) and so is the pair $(\hat{\mu}, \hat{b})$. By contradiction, assume that $\hat{\mu}$ is not optimal for Problem (4). Then there exists a feasible $\mu^\dagger$ for (4) with $sse_{\mathcal{C}}(\mu^\dagger) < sse_{\mathcal{C}}(\hat{\mu})$. Via

reconstructing the ML constrained Problem (2), we obtain $sse_{\hat{\mathcal{C}}}(\mu^\dagger, b^\dagger) = sse_{\mathcal{C}}(\mu^\dagger) + C < sse_{\mathcal{C}}(\hat{\mu}) + C = sse_{\hat{\mathcal{C}}}(\mu^*, b^*)$, which contradicts the optimality of $(\mu^*, b^*)$. Hence, the optimal solution for Problem (2) is optimal for Problem (4). $\qquad\square$

## C   CONVERGENCE ANALYSIS

In this section we establish the convergence of the proposed BB scheme, constructed with the grouping–based Lagrangian decomposition lower bound and the decompsable upper bound based on closed form solutions for must-link or K-coloring. A key feature of our algorithm is that it **branches exclusively in the space of first-stage variables** $\mu$ **to guarantee convergence**. As all must-link components have been collapsed and every CL-infeasible assignment eliminated, the remaining problem is an *unconstrained* optimization over $\mu$ with continuous objective $Q(\mu) = \sum_{s \in \hat{\mathcal{S}}} Q_s(\mu)$.

Therefore, the proof of convergence can easily adopt the foundational results of (Cao & Zavala, 2019) and the seminal contributions in Chapter IV of (Horst & Tuy, 2013). Although the original pairwise–constrained MSSC places additional feasibility requirements on the assignment variables, our *equivalent unconstrained* re-formulation—obtained by first collapsing every must-link component (Theorem 3.4) and then discarding all assignments that violate cannot-link constraints (See Lemma 3.1 and 3.2)—allows *any* point $\mu \in M$ to be treated as a feasible first-stage decision. The proof of Theorem 3.5 is thus becoming obvious with the definitions and theoretical frameworks of Cao & Zavala (2019), while only notational adaptations will be processed to reflect the reduced dataset $\hat{\mathcal{S}}$ and the viable-cluster sets $\{\mathcal{K}_s\}_{s \in \hat{\mathcal{S}}}$ specific to the present problem.

**Lemma C.1** (Lower Bounding Consistency). *Given an exhaustive subdivision (See Definition IV.10 (Horst & Tuy, 2013)) on $\mu$, the lower-bounding operation in Algorithm 1 is strongly consistent (See Definition IV.7 (Horst & Tuy, 2013)).*

*Proof.* With an exhaustive subdivision, each box $M_{i_q}$ shrinks to a single point $\bar{\mu}$, so $\bar{M} = \{\bar{\mu}\}$. We prove that $\lim_{q \to \infty} \beta(M_{i_q}) = z(\bar{M}) = \sum_{s \in \hat{\mathcal{S}}} Q_s(\bar{\mu})$. Define, for every sample $s$, $\tilde{\mu}_{i_q,s} \in \arg\min_{\mu \in M_{i_q}} \min_{k \in \mathcal{K}_s} \|x_s - \mu_k\|_2^2$, where $\mathcal{K}_s$ is the set of clusters still admissible for $s$ after the cannot-link pruning. **Because each $\mathcal{K}_s$ already excludes every cannot-link pairing, every distance minimized in the definition of $\tilde{\mu}_{i_q,s}$ automatically respects all CL constraints.** Since $M_{i_q} \to \{\bar{\mu}\}$, we have $\tilde{\mu}_{i_q,s} \to \bar{\mu}$. Using the continuity of $Q_s(\cdot)$, it follows that $Q_s(\bar{\mu}) = \lim_{q \to \infty} Q_s(\tilde{\mu}_{i_q,s}) = \lim_{q \to \infty} \beta_s(M_{i_q})$. Summing over all $s$ yields $\lim_{q \to \infty} \beta(M_{i_q}) = \sum_{s \in \hat{\mathcal{S}}} Q_s(\bar{\mu})$. Proof complete. $\quad\square$

**Lemma C.2** (Lower Bounding Convergence). *Given an exhaustive subdivision (Definition IV.10 (Horst & Tuy, 2013)) on $\mu$, Algorithm 1 satisfies $\lim_{i \to \infty} \beta_i = z$.*

*Proof.* This result can be obtained from Lemma C.1 and Theorem IV.3 of (Horst & Tuy, 2013). $\quad\square$

**Lemma C.3** (Upper Bounding Convergence). *Given an exhaustive subdivision (Definition IV.10 (Horst & Tuy, 2013)) on the centroid space $\mu$, Algorithm 1 produces a sequence $\{\alpha_i\}$ that satisfies $\lim_{i \to \infty} \alpha_i = z$.*

*Proof.* Let $\mu^* \in M_0$ be an optimal centroid set for the *equivalent unconstrained* MSSC obtained after **collapsing must-link components and discarding every cannot-link–infeasible assignment**. According to Lemma 6 in Cao & Zavala (2019), $\lim_{i \to \infty} \alpha_i = z$ holds when executing Algorithm 1.

$\square$

Combing Lemma C.2 and C.3, we obtain Theorem 3.5. Essentially, pseudo-samples in Section 3.2 yield an equivalent unconstrained MSSC problem, and Theorem 3.4 proves a bijection between optimal solutions before and after this transformation. The geometric rules in Section 3.1 apply dominance checks via Lemmas 3.1 and 3.2, excluding only assignments whose cost significantly exceeds the incumbent upper bound without removing any centroid regions. Consequently, our branch-and-bound algorithm still exhaustively subdivides the centroid space, satisfying Lemmas C.1 through C.3. Thus, the result remains valid for the entire SDC-GBB pipeline. Empirically, we show that within a fixed 12-hour runtime limit, SDC-GBB achieves optimality gaps above 0.1% for some datasets under certain constraints. However, this does not reflect the failure of convergence in our

global optimization scheme, but rather the best feasible solution and its lower bound at timeout. Imposing a time limit is the standard which we follow to ensure a fair comparison with other exact baselines.

# D EFFECT OF PAIRWISE CONSTRAINTS

Background knowledge in constrained clustering is introduced through pairwise constraints: must-link (ML) and cannot-link (CL). This section analyzes how varying densities of these constraints affect the branch-and-bound performance on medium-sized problem instances. Table 4 summarizes the number of nodes processed and the average computational time per node under different constraint densities. All experiments reported in Table 4 achieve a final optimality gap below or equal to $0.1\%$.

Must-link constraints merge linked samples into single pseudo-points before initiating the branch-and-bound procedure. Conceptually, this operation is analogous to samples merging immediately upon defining constraints, effectively knowing in advance that they must converge into a single optimal position. Figure 2 illustrates this merging process: each must-link pair collapses into one pseudo-point, thus reducing the number of samples to consider. Although this merging simplifies the optimization search space by reducing dimensionality, it simultaneously imposes additional equality constraints in each node relaxation within the branch-and-bound process, thereby increasing the computational effort per node relaxation. Nonetheless, overall node processing becomes faster since fewer distinct samples remain active, which accelerates bound computations without compromising the equivalence and optimality of the final solution. Empirically, the average time per node decreases with an increasing number of must-link constraints (from 74 seconds per node down to 13 seconds per node for dataset *Syn-2100*). However, there is a threshold for the density of constraints necessary to significantly simplify the branching process: below approximately $n/64$ must-link pairs, constraints have minimal practical relevance, causing the equivalent problem formulation, described in Section 3.2, to behave similarly to its unconstrained counterpart.

In contrast, the geometric sample-determination strategy for cannot-link constraints functions as barriers that restrict feasible assignments, similar to placing walls within an axis-aligned region. Unlike must-link constraints, which collapse samples proactively, cannot-link constraints compel each sample's assignment to navigate around imposed boundaries that are not initially evident. Each sample seeks to reach its optimal cluster centroid but must repeatedly avoid these geometric barriers, reflecting constrained assignments and generating additional branching iterations. Despite this growth in node count, the computational time per node remains stable because infeasible assignments are pruned at an early stage, as stated in Lemma 3.1. Thus, the complexity introduced by cannot-link constraints primarily affects the extent of branching rather than the computational complexity of each bound evaluation. In summary, must-link constraints simplify the optimization upfront by reducing per-node complexity through component collapse, whereas cannot-link constraints expand the search tree but maintain per-node computational cost, ensuring that solutions satisfy the imposed constraints without fundamentally altering the underlying clustering structure.

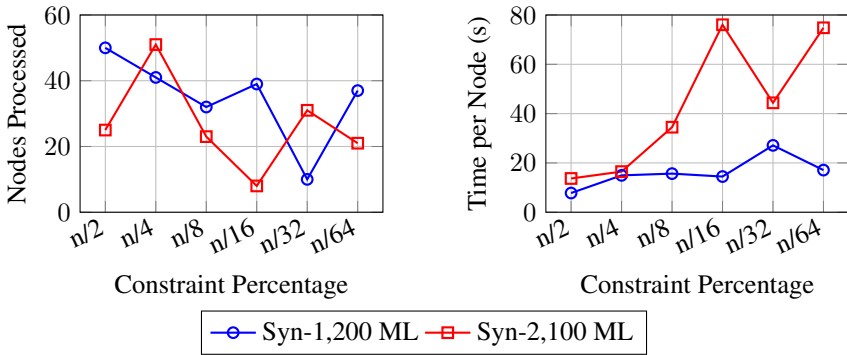

Figure 3: Effect of Must–Link Constraints on (a) the number of nodes processed and (b) time per node, for both synthetic datasets.

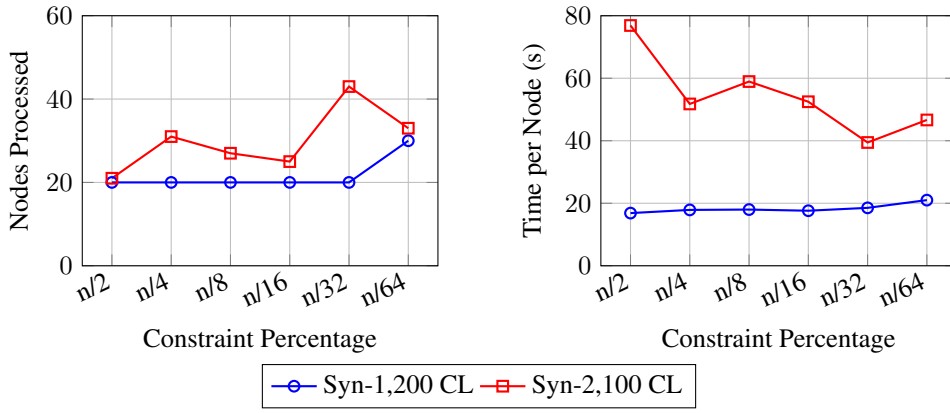

Figure 4: Effect of Cannot–Link Constraints on (a) the number of nodes processed and (b) time per node, for both synthetic datasets.

Interestingly, we observe that the linear relaxation is substantially weakened when must-link constraints form large or overlapping superpoints. In URBANGB, the merging of 469 groups into $k = 3$ superpoints induces symmetry that leaves a 12.97% gap, and in SPNET3D the near one-dimensional data arrangement produces overlapping superpoints that the bound cannot improve, resulting in a 7.43% gap. On the other hand, in SKIN_8 and SPNET3D_5, the mixed ML+CL case is uniquely hard because ML contraction creates component-level "hotspots" that are densely entangled by cannot-links. Inside SKIN_8, there is a massive ML super-component that must occupy one cluster and is CL-forbidden from sharing with thousands of neighbors, whereas the tight CL and loose ML constraints simultaneously oppose the geometry of SPNET3D_5. This coupling weakens the relaxation and drives deep branching, yielding a large optimality gap > 3% in the mixed constraint case, whereas ML-only or CL-only avoid this interaction and remain tight.

Table 4: Runtime metrics with Different Constraint Settings with solution optimality gap < 0.1%.

| DATASET | METRIC | $\frac{n}{2}$ | $\frac{n}{4}$ | $\frac{n}{8}$ | $\frac{n}{16}$ | $\frac{n}{32}$ | $\frac{n}{64}$ |
|---------|--------|---------------|---------------|---------------|----------------|----------------|----------------|
| | | **MUST-LINK (ML)** | | | | | |
| Syn-1,200 | NODES | 50 | 41 | 32 | 39 | 10 | 37 |
| | GAP (%) | <0.1% | <0.1% | <0.1% | <0.1% | <0.1% | <0.1% |
| | TIME (S) | 389.61 | 613.09 | 501.05 | 564.41 | 271.24 | 632.25 |
| | TIME/NODE (S) | 7.79 | 14.95 | 15.66 | 14.47 | 27.12 | 17.09 |
| | CORE HOUR (H) | 7.79 | 14.95 | 15.66 | 14.47 | 27.12 | 17.09 |
| Syn-2,100 | NODES | 25 | 51 | 23 | 8 | 31 | 21 |
| | GAP (%) | <0.1% | <0.1% | <0.1% | <0.1% | <0.1% | <0.1% |
| | TIME (S) | 342.20 | 840.70 | 793.19 | 608.32 | 1375.86 | 1570.95 |
| | TIME/NODE (S) | 13.69 | 16.48 | 34.49 | 76.04 | 44.38 | 74.81 |
| | CORE HOUR (H) | 5.41 | 8.52 | 6.96 | 7.84 | 3.77 | 8.78 |
| | | **CANNOT-LINK (CL)** | | | | | |
| Syn-1,200 | NODES | 20 | 20 | 20 | 20 | 20 | 30 |
| | GAP (%) | <0.1% | <0.1% | <0.1% | <0.1% | <0.1% | <0.1% |
| | TIME (S) | 336.98 | 357.45 | 359.61 | 352.23 | 370.58 | 629.90 |
| | TIME/NODE (S) | 16.85 | 17.87 | 17.98 | 17.61 | 18.53 | 21.00 |
| | CORE HOUR (H) | 7.79 | 14.95 | 15.66 | 14.47 | 27.12 | 17.09 |
| Syn-2,100 | NODES | 21 | 31 | 27 | 25 | 43 | 33 |
| | GAP (%) | <0.1% | <0.1% | <0.1% | <0.1% | <0.1% | <0.1% |
| | TIME (S) | 1614.36 | 1604.81 | 1591.37 | 1312.08 | 1695.81 | 1539.55 |
| | TIME/NODE (S) | 76.87 | 51.77 | 58.94 | 52.48 | 39.44 | 46.65 |
| | CORE HOUR (H) | 7.79 | 14.95 | 15.66 | 14.47 | 27.12 | 17.09 |

To further highlight resource usage with a unified measure of parallelism and wall-clock time, we compute "core-hour", defined as $core\_hour = \frac{time(s) \times cores}{3600(s)}$. Under must-link constraints, SDC-GBB's overall branching behavior varies substantially with constraint density, as on Syn-1,200 and Syn-2,100, going from $n/2$ to $n/32$ ML constraints generally reduces the number of branch-and-bound nodes but raises the average time per node. In contrast, the number of nodes explored grows roughly linearly under uniform cannot-link constraint placement; as we observe in Table 4, node count scales $\sim O(m)$ with $m$ cannot-link constraints, while time per node remains constant. As a result, the wall-clock time increases by about 20% on Syn-1,200 and roughly a factor of two on Syn-2,100, but the core hour stays relatively static across constraint sizes.

When connecting this empirical measure to asymptotic complexity, we note that each branch-and-bound node incurs $O(|\hat{S}|K)$ work, while exhaustive axis-aligned bisection yields at most $O((\delta/\varepsilon)^{Km})$ (Horst & Tuy, 2013), where $\delta$ is the initial box size and $\varepsilon$ the target precision. Together these give a nominal worst-case cost of $O((\delta/\varepsilon)^{Km}|\hat{S}|K)$. On a cluster with $P$ identical CPU cores, parallelising across open nodes yields an expected core-hour consumption of $O\left(\frac{(\delta/\varepsilon)^{Km}|\hat{S}|K}{P}\right)$, provided the number of simultaneously available nodes exceeds $P$. The empirical trends observed in Tables 1 and 2 align with this theoretical envelope.

### D.1    RUNTIME ANALYSIS OF EXHAUSTIVE $\mu$ SEARCH

Table 5 reports wall-clock statistics for the exhaustive enumeration of $\mu$ values inside our reduced-space branch–and–bound framework [2]. For each synthetic dataset we list the total wall time ($T_{\text{total}}$), the time devoted exclusively to the $\mu$ search ($T_\mu$), the number of branch–and–bound nodes explored ($N_{\text{nodes}}$), and the average time per node ($T_{\text{node}} = T_\mu/N_{\text{nodes}}$).

Table 5: Runtime metrics for n/4 must-links exhaustive $\mu$ search (gap $\leq 0.1\%$).

| DATASET | $T_{\text{TOTAL}}$ (S) | $T_\mu$ (S) | $N_{\text{NODES}}$ | $T_{\text{NODE}}$ (S) |
|---|---|---|---|---|
| SYN-21000 | 3,323.69 | 3,311.58 | 69 | 47.99 |
| SYN-81000 | 12,921.19 | 12,804.11 | 71 | 180.34 |
| SYN-141000 | 18,499.32 | 18,307.11 | 55 | 332.86 |
| SYN-171000 | 28,796.98 | 28,551.09 | 43 | 664.00 |

The results confirm that the $\mu$ enumeration dominates the computational budget ($T_\mu/T_{\text{total}} > 0.99$), while other tasks, namely relaxations, cuts and I/O are marginal. Although $T_{\text{node}}$ grows faster than linearly with the dataset size, parallel execution on 100 cores keeps the overall wall-time below ten hours even for the 171 k-point instance.

### D.2    CLUSTERING EVALUATION

Our work distinguishes between two aspects of clustering performance: (i) the formulation, which defines how data similarity is measured and affects metrics such as ARI, NMI, and purity; and (ii) the quality of the solution (in terms of cost minimization), which we measure using the optimality gap. Our focus is on the quality of the solution for the K-Means cost, ensuring an optimality guarantee of the solution. Here, we performed additional statistical tests and gave the ARI, NMI and purity results of 5 datasets with ground truth labels [3] and compare these with the algorithm of (Hua et al., 2021) in Table 6 [4]. In these experiments, the number of clusters is set to the number of ground truth labels.

---

[2]Node count decreases with $n$ because larger ML components reduce $|\hat{S}|$ via pseudo-sample collapse (Section 3.2), enabling scalability up to 1.5M samples under ML constraints.

[3]ARI/NMI/Purity are reported only as external validation, while solution quality is measured via certified SSE gaps.

[4]Identical metrics confirm that CL constraints are satisfied at the global optimum; enforcement via Lemmas 3.1 –3.2 and node feasibility checks remains active throughout the tree.

Table 6: Clustering evaluation metrics on solutions of datasets under different constraint settings.

| Metrics | Constraints | Iris $k = 3$ | Seeds $k = 3$ | Hemi $k = 3$ | HTRU2 $k = 2$ | Skin_8 $k = 2$ |
|---|---|---|---|---|---|---|
| ARI | MSSC (Hua et al., 2021) | 0.7163 | 0.7166 | 0.0126 | $-0.0779$ | $-0.0387$ |
| | SDC-GBB (ML) | 0.7859 | 0.7261 | 0.0137 | $-0.0385$ | $-0.0427$ |
| | SDC-GBB (CL) | 0.7163 | 0.7166 | 0.0148 | 0.0389 | 0.3545 |
| | SDC-GBB (ML+CL) | 0.7711 | 0.7384 | 0.0171 | 0.0909 | $-0.0090$ |
| NMI | MSSC (Hua et al., 2021) | 0.7419 | 0.6949 | 0.0335 | 0.0265 | 0.0221 |
| | SDC-GBB (ML) | 0.7773 | 0.6979 | 0.0335 | 0.0666 | 0.0280 |
| | SDC-GBB (CL) | 0.7419 | 0.6949 | 0.0309 | 0.1007 | 0.4388 |
| | SDC-GBB (ML+CL) | 0.7705 | 0.7006 | 0.0338 | 0.1275 | 0.0343 |
| Purity | MSSC (Hua et al., 2021) | 0.8867 | 0.8952 | 0.4210 | 0.9084 | 0.7925 |
| | SDC-GBB (ML) | 0.9200 | 0.9000 | 0.4251 | 0.9084 | 0.7925 |
| | SDC-GBB (CL) | 0.8867 | 0.8952 | 0.4373 | 0.9084 | 0.9420 |
| | SDC-GBB (ML+CL) | 0.9133 | 0.9048 | 0.4419 | 0.9084 | 0.7925 |

# E    Heuristic Algorithms

We evaluate the proposed procedure by comparing its clustering quality with four reference heuristics for the minimum-sum-of-squares clustering problem subject to must-link (ML) and cannot-link (CL) constraints. COP-$k$-means (Wagstaff et al., 2001) restarts the classical Lloyd algorithm one hundred times and enforces the constraints at every assignment step. The post-processing encode-$k$-means-Post method of Nghiem (Nghiem et al., 2020) formulates the reassignment of instances produced by an unconstrained or partially constrained clustering method as a binary combinatorial program that respects all ML and CL relations. The binary linear programming approach of Baumann (Baumann, 2020) (BLPKM-CC) solves to optimality the assignment subproblem within each Lloyd iteration; only the initial centroids are random, so the method is partially deterministic. Variants of coreset algorithm construct a $(k, \epsilon)$-coreset, on which $k$-means can be solved quickly while guaranteeing that the resulting centers incur at most a $(1 + \epsilon)$ multiplicative error in squared-error cost on the full dataset, thus preserving near-optimality with far lower computational and memory demands. We test several coreset construction algorithms and obtain the UB for Sensitivity Sampling with $k = 3$, oversample factor $c = 2$, error $\epsilon = 0.1$ and probability of approximation guarantee $delta = 0.1$, as this is the state-of-the-art method for constructing coresets (Schwiegelshohn & Sheikh-Omar, 2022) and the only method satisfying the 4-hour runtime limit for all datasets.

Tables 7, 8, 9 report the optimal UB obtained by all heuristic algorithms with 100 independent initializations for ML-only, CL-only, and ML+CL experiments respectively. We do not include results for Sensitivity Sampling in experiments involving CL since under coreset algorithms, any hard cannot-link requirement collapses the additivity assumption and blows up point sensitivities, thus inflates the coreset to linear size. Besides, we apply N/A to some COP-$k$-means results, as this algorithm generally could not find the global optima for datasets of size $n > 2,000$.

Table 7: Heuristic algorithms on $\frac{n}{4}$ ML constraints

| DATASETS | SIZE | COP-$k$-MEANS | ENCODE-$k$-MEANS-POST | BLPKM-CC | SENSITIVITY SAMPLING |
|---|---|---|---|---|---|
| IRIS | 150 | 150.78 | 84.67 | **83.82** | 93.87 |
| SEEDS | 200 | 713.88 | 625.37 | **620.78** | 761.60 |
| HEMI | 1,955 | N/A | $\mathbf{1.602 \times 10^7}$ | $1.875 \times 10^7$ | $2.167 \times 10^7$ |
| PR2392 | 2,392 | N/A | $\mathbf{3.210 \times 10^{10}}$ | $3.246 \times 10^{10}$ | $3.436 \times 10^{10}$ |
| RDS_CNT | 10,000 | N/A | $\mathbf{6.122 \times 10^7}$ | $6.579 \times 10^7$ | $6.387 \times 10^7$ |
| HTRU2 | 17,898 | N/A | $\mathbf{1.407 \times 10^8}$ | $1.472 \times 10^8$ | $1.505 \times 10^8$ |
| SPNET3D_5 | 50,000 | N/A | $\mathbf{6.627 \times 10^6}$ | $7.089 \times 10^6$ | $7.196 \times 10^6$ |
| SKIN_8 | 80,000 | N/A | $5.464 \times 10^8$ | $5.492 \times 10^8$ | $\mathbf{4.533 \times 10^8}$ |
| URBANGB | 360,177 | N/A | OUT OF MEMORY | | $\mathbf{1.643 \times 10^9}$ |
| SPNET3D | 434,874 | N/A | $\mathbf{5.848 \times 10^7}$ | $6.264 \times 10^7$ | $6.413 \times 10^7$ |
| SYN-42000 | 42,000 | N/A | $\mathbf{5.127 \times 10^5}$ | $1.584 \times 10^6$ | $5.148 \times 10^5$ |
| SYN-210000 | 210,000 | N/A | $\mathbf{2.163 \times 10^6}$ | $\mathbf{2.163 \times 10^6}$ | $2.178 \times 10^6$ |
| SYN-420000 | 420,000 | N/A | $\mathbf{6.010 \times 10^6}$ | $\mathbf{6.010 \times 10^6}$ | $6.080 \times 10^6$ |
| SYN-1050000 | 1,050,000 | N/A | $3.708 \times 10^7$ | $\mathbf{1.050 \times 10^7}$ | $1.052 \times 10^7$ |
| SYN-1500000 | 1,500,000 | N/A | $5.611 \times 10^7$ | $\mathbf{1.725 \times 10^7}$ | $1.731 \times 10^7$ |

Table 8: Heuristic algorithms on $\frac{n}{4}$ CL constraints

| DATASET | SIZE | COP-$k$-MEANS | ENCODE-$k$-MEANS-POST | BLPKM-CC |
|---|---|---|---|---|
| IRIS | 150 | 119.37 | **80.31** | 80.71 |
| SEEDS | 200 | 634.3 | 603.96 | **603.04** |
| HEMI | 1,955 | $1.711 \times 10^7$ | $\mathbf{1.401 \times 10^7}$ | $1.606 \times 10^7$ |
| PR2392 | 2,392 | $2.596 \times 10^{10}$ | $\mathbf{2.566 \times 10^{10}}$ | $2.578 \times 10^{10}$ |
| RDS_CNT | 10,000 | $3.696 \times 10^7$ | $\mathbf{2.897 \times 10^7}$ | $2.902 \times 10^7$ |
| HTRU2 | 17,898 | N/A | $1.928 \times 10^8$ | $\mathbf{1.740 \times 10^8}$ |
| SPNET3D_5 | 50,000 | N/A | $\mathbf{3.938 \times 10^6}$ | $4.027 \times 10^6$ |
| SKIN_8 | 80,000 | N/A | $\mathbf{6.367 \times 10^8}$ | $6.464 \times 10^8$ |
| SYN-12000 | 12,000 | N/A | $\mathbf{9.503 \times 10^4}$ | $\mathbf{9.503 \times 10^4}$ |
| SYN-21000 | 21,000 | N/A | $1.928 \times 10^8$ | $\mathbf{1.740 \times 10^8}$ |
| SYN-42000 | 42,000 | N/A | $\mathbf{1.817 \times 10^5}$ | $\mathbf{1.817 \times 10^5}$ |
| SYN-210000 | 210,000 | N/A | $\mathbf{2.161 \times 10^6}$ | $\mathbf{2.161 \times 10^6}$ |

Table 9: Heuristic algorithms on $\frac{n}{4}$ ML + $\frac{n}{4}$ CL constraints

| DATASET | SIZE | COP-$k$-MEANS | ENCODE-$k$-MEANS-POST | BLPKM-CC |
|---|---|---|---|---|
| IRIS | 150 | N/A | 88.75 | **86.85** |
| SEEDS | 200 | N/A | 601.36 | **597.14** |
| HEMI | 1,955 | N/A | $\mathbf{1.566 \times 10^7}$ | $1.762 \times 10^7$ |
| PR2392 | 2,392 | N/A | $\mathbf{2.922 \times 10^{10}}$ | $2.945 \times 10^{10}$ |
| RDS_CNT | 10,000 | N/A | $\mathbf{7.579 \times 10^7}$ | $7.919 \times 10^7$ |
| HTRU2 | 17,898 | N/A | $2.218 \times 10^8$ | $\mathbf{1.859 \times 10^8}$ |
| SPNET3D_5 | 50,000 | N/A | $\mathbf{8.171 \times 10^6}$ | $8.320 \times 10^6$ |
| SKIN_8 | 80,000 | N/A | $\mathbf{7.579 \times 10^8}$ | $8.775 \times 10^8$ |
| SYN-12000 | 12,000 | N/A | $\mathbf{9.520 \times 10^4}$ | $\mathbf{9.520 \times 10^4}$ |
| SYN-21000 | 21,000 | N/A | $\mathbf{1.818 \times 10^5}$ | $\mathbf{1.818 \times 10^5}$ |
| SYN-42000 | 42,000 | N/A | $\mathbf{5.133 \times 10^5}$ | $\mathbf{5.133 \times 10^5}$ |
| SYN-210000 | 210,000 | N/A | $\mathbf{2.165 \times 10^6}$ | $\mathbf{2.165 \times 10^6}$ |

