# OpenReview forum: "A Scalable Global Optimization Algorithm For Constrained Clustering"
_ICLR.cc/2026/Conference — Submitted to ICLR 2026_

### Official Review · Reviewer_2GvY · 2025-10-23

**Soundness:** 3
**Presentation:** 2
**Contribution:** 3
**Rating:** 4
**Confidence:** 3

**Summary:**

This paper presents a new algorithm for optimally solving sum of squared errors clustering problems (k-means) in the constrained setting, where one includes certain must-link and cannot-link constraints on certain pairs of points. Since this is an NP-hard problem, optimal solutions for it are hard to compute and existing methods do not scale to very large datasets.

The new method proposed in this paper scales to much larger datasets than existing techniques, reaching up to instances with 1.5 million samples for must-link constrained cases and 200k samples for cannot link cases, with low optimality gaps. Many prior techniques struggle to obtain optimal solutions with hundreds of points, and the best existing results, according to this paper, only scale to instances around 1000 points.

The paper achieves these speed-ups by introducing geometric sample determination rules to eliminate cannot-links, and merge must link constraints into pseudosamples. These tricks speed up the branch and bound process for finding optimal solutions; the BB approach specifically branches on cluster-center variables rather than sample-to-cluster variables.

**Strengths:**

* The constrained clustering problem is well-motivated and is a very relevant topic; it is interesting to develop exact algorithms for this problem even though it is NP-hard
* The numerical experiments seem impressive and do appear to significantly improve in practice over previous techniques.
* The paper incorporates helpful figures in presenting the theoretical results.
* The structure of the paper is fairly good. There do not appear to be many issues with typos or grammar

**Weaknesses:**

There are two weaknesses:

W1: The key contribution in section 3.1 is unclear due to technical writing issues

In the Sections 2-3, there are many technical details that are not explained precisely to the point where the core technical approach is unclear to me. I am not intimately familiar with research on branch-and-bound algorithms for clustering, but I have good general familiarity with optimization methods for clustering problems, and there were many technical details I could not parse, often because notation was not properly explained or defined.

Here are some questions I had in the technical details while reading specifically through Section 2 and Section 3.1

* Is Problem 2 supposed to be an equivalent reformulation of Problem 1? I'm pretty sure it is, but that's not explicitly stated, and you call it a semi-supervised MSSC, which is the first time you use that term (you did not use it to describe Problem 1) so at first it's not clear if you're introducing some equivalent formulation or a new problem. It's also not entirely clear to me what Problem 2 has to do with what is in section 3
* In Problem 2, are you missing a "for all k \in \mathcal{K}" in constraints b and c?
* What's a big-M constant? (line 121)
* In line 139, it's not clear to me why \rho is the current incumbent cost. The objective function is a sum of squares, whereas \rho is the maximum distance between a point and a centroid. These are not the same thing. I'm not sure what I'm missing, but this doesn't seem explained very clearly.
* In line 145, you refer to (s,k) as a candidate pair, but you don't explain what that means and it's not immediately clear what you're saying.
* In the definition of \rho, you use the notation \mu_{k(s)}^\text{best}. This is used without a clear explanation or definition before being used. You do explain right afterward that this is the "centroid to which [a sample] is assigned in the current best solution", which is good, but it'd be helpful to have a clearer explanation beforehand including what k(s) represents.
* You say M_k represents any region, but then specify you mean an axis aligned box. "region" is a much more general term than "axis-aligned box" though, so I don't think you really mean "any" region.
* The notation for M_k seems very close to M_0, which is used to mean something completely different later (the "solution space"). This is a bit confusing.
* In some places you write d_{min}(x_s, M_k), and in the figures you use d_{min}^i for some integer i. Can you give a more specific mapping between these notations so a reader can parse it more readily?

Obviously, some of these issues are larger than others. But when taken all together, I ultimately was not able to follow the core contribution in Section 3.1 and what Lemmas 3.1 and 3.2 are ruling out.

Weakness 2: The technical contribution in 3.2 seems quite limited.

In particular, this section seems to just be making the observation that if you include a must-link constraint, you can just collapse points into a new sample and then solving an equivalent unconstrained problem. This is a pretty basic and standard observation in constrained clustering, and it's significance is maybe a bit overstated. In the summary of contributions in the introductions this is described as "We introduce a centroid-based pseudosample formulation for must-link subsets", but that seems like an overly complicated way to describe the simple thing going on.

One final minor comment: the opening sentence in 3.3 is identical to the opening sentence in 3.4. Was this intended? It seems very repetitive.

**Questions:**

Can you please clarify my questions regarding the notation and points being made, especially in section 3.1?

---

> ### Author Response · Authors · 2025-11-17
> **Response to Reviewer 2GvY (1/3)**
>
> **The core scope of our work is to advance the scalability of deterministic global optimization for constraint clustering**, enabling high-quality, reproducible solutions on datasets orders of magnitude larger than previously possible by exact methods.
>
> > **Q1.1**: Is Problem 2 supposed to be an equivalent reformulation of Problem 1? I'm pretty sure it is, but that's not explicitly stated, and you call it a semi-supervised MSSC (not used for Problem 1), so at first it's not clear if you're introducing an equivalent formulation or a new problem.
>
> Yes. **Problem (2) is an exact mixed-integer second-order cone program (MISOCP) reformulation of Problem (1) and both encode the same semi-supervised MSSC with pairwise ML/CL constraints**.
>
> Problem (1) states the pairwise-constrained MSSC directly in terms of assignment variables $b_{s,k}$ and centroids $\mu_k$. Problem (2) reuses the same ML/CL and assignment structure and introduces only the auxiliary distance variables $d_{s,k}$ and $d_{s,*}$ and big-$M$ linking constraints (2b)--(2c), so that **the model becomes a MISOCP amenable to conic solvers and to our bounding scheme**.
>
> Thus, Problems (1) and (2) have the same feasible set and the same optimal value; Problem (2) does not define a different task; it is a conic reformulation of the original pairwise-constrained MSSC.
>
> **For the camera-ready revision**, we will add a statement:
> “Problem (2) is a MISOCP reformulation of Problem (1); both are equivalent formulations of the same semi-supervised MSSC with pairwise ML/CL constraints,”
> and we will consistently refer to Problem (1) as a semi-supervised MSSC as well.
>
> ---
>
> > **Q1.2**: It's also not entirely clear to me what Problem 2 has to do with what is in section 3.
>
> **Problem (2) is the mathematical backbone of Section 3: the reduced-space branch-and-bound algorithm is derived from its MISOCP structure**.
>
> First, the opening of Section 3 states that the algorithm is built on the MISOCP in Section 2:
>
> > > *“We tailor this scheme to the pairwise-constrained clustering problem by integrating geometric probing rules **derived from the MISOCP formulation in Sec. 2** to tighten both lower and upper bounds.”* (Section 3, pp. 3)
>
> Second, **Section 3.1** turns the conic constraints of Problem (2) into geometric sample-determination rules. Using the incumbent cost $\rho$ and the distance bounds $d_{\min}(x_s, M_k)$ and $d_{\max}(x_s, M_k)$, the manuscript notes that
>
> > > *"because $\rho$ is an upper bound on the true assignment cost, any candidate pair $(s,k)$ with $d_{\min}(x_s, M_k) > \rho$ can never be optimal."* (Section 3.1, pp. 3).
>
> which directly leads to Lemmas 3.1--3.2. These lemmas are geometric restatements of the assignment and distance structure encoded in Problem (2).
>
> Third, **Section 3.2 explicitly builds on Problem (2)**. The subsection *“Equivalent Unconstrained Clustering Problem”* begins with:
>
> > > *“To isolate this effect, consider the **ML-only version of Problem (2)** ”* (Section 3.2, pp. 4)
>
> We then show that collapsing each must-link component into pseudo-samples yields an unconstrained instance with the **same global optimum** as the ML-only MISOCP (Theorem 3.4). This correspondence allows Sections 3.3–3.4 to run reduced-space BB on the aggregated instance while still certifying global optimality for the original ML-constrained MSSC.
>
> **For the camera-ready revision**, we will make this dependency even more explicit by adding cross-references in Section 3 (“derived from the MISOCP formulation (2)” in the geometric rules and pseudo-sample construction), but Section 3 is built on Problem (2), not independent.
>
> ---
>
> > **Q2**: In problem 2, are you missing a "for all $k \in \mathcal{K}$" in constraints b and c?
>
> Constraints (2b)–(2c) hold **for every** sample–cluster pair and should be read as
> $$
> -N(1-b_{s,k}) \le d_{s,*} - d_{s,k} \le N(1-b_{s,k}),\quad
> d_{s,k} \ge |x_s - \mu_k|_2^2,\quad \forall s \in \mathcal{S},\ \forall k \in \mathcal{K}.
> $$
>
> **For the camera-ready revision**, we will make this explicit in the revised manuscript by adding the quantifiers “$(\forall s \in \mathcal{S}, \forall k \in \mathcal{K})$” to constraints (2b)–(2c).
>
> ---
>
> > **Q3**: What's a big-M constant?
>
> A big-$M$ constant is a large upper bound used in mixed-integer optimization so that, when a binary variable takes value 0, the associated constraint becomes effectively inactive; when the binary is 1, the constraint is enforced in its tight form.
>
> In Appendix A, we define a uniform big-$M$ bound over all samples and clusters as
> $$
> N = \max_{s,k} \sum_{i=1}^m \max\Bigl(\bigl|x_{s,i} - \mu_{k,i}^l\bigr|^2,\ \bigl|x_{s,i} - \mu_{k,i}^u\bigr|^2\Bigr),
> $$
> computed at the root node from the extreme data coordinates and centroid bounds $(\mu^l,\mu^u)$. This choice ensures $N$ dominates all possible squared distances $|x_s - \mu_k|_2^2$ within the bounding box $[\mu^l,\mu^u]$ and is therefore a **valid and tight** big-$M$ constant for constraints (2b)–(2c).

---

> ### Author Response · Authors · 2025-11-17
> **Response to Reviewer 2GvY (2/3)**
>
> > **Q4**: In line 139, it's not clear to me why $\rho$ is the current incumbent cost. The objective function is a sum of squares, whereas $\rho$ is the maximum distance between a point and a centroid. These are not the same thing. I'm not sure what I'm missing, but this doesn't seem explained very clearly.
>
> In our notation, the **incumbent objective value** is the full MSSC cost
> $$
> z^{\text{inc}} := \sum_{s\in\mathcal S} \bigl|x_s - \mu^{\text{best}}_{k(s)}\bigr|_2^2.
> $$
>
> For each sample we denote its **per-sample contribution** as
> $ \rho_s := \|x_s - \mu^{\text{best}}_{k(s)}\|_2^2 $
> , and define
>
> $ \rho := \max_{s\in\mathcal S}\, \rho_s $.
>
> So $\rho$ is **not** the full objective value; it is the **worst per-sample squared distance** in the current incumbent solution, and therefore a **per-sample upper bound** on the true assignment cost:
> $$
> \forall s\in\mathcal S:\quad \bigl|x_s - \mu^{\text{best}}_{k(s)}\bigr|_2^2 \le \rho \le z^{\text{inc}}.
> $$
>
> Lemma 3.1 then uses $\rho$ precisely as this per-sample threshold. For any node and any region $M_k$,
> $$
> d_{\min}(x_s, M_k) > \rho
> $$
>
> implies that **even in the best case** (choosing the most favorable $\mu \in M_k$), assigning sample $s$ to cluster $k$ would incur a cost strictly larger than its worst incumbent cost contribution. Hence no solution with total cost $\le z^{\text{inc}}$ can have $b_{s,k}=1$, and we can safely fix $b_{s,k}=0$ in every optimal solution with objective value no larger than the incumbent.
>
> **For the camera-ready revision,** we will clarify this in the manuscript by replacing the sentence “Thus, $\rho$ represents the current incumbent cost” with a more precise statement such as: *“Thus, $\rho$ represents the **maximum per-sample contribution** to the current incumbent cost and is used as a per-sample upper bound in the geometric elimination rules.”*
>
> ---
>
> > **Q5**: In line 145, you refer to (s, k) as a candidate pair, but you don't explain what that means and it's not immediately clear what you're saying.
>
> We agree that the notion of candidate pair should be stated explicitly. In our notation, a *candidate pair* $(s,k)$ is any sample–cluster pair whose assignment has not yet been ruled out at the current branch-and-bound node; that is, a point $s$ and cluster $k$ such that assigning $s$ to $k$ remains feasible under the current geometric tests and must-link/cannot-link (ML/CL) constraints.
>
> > **Q6**: It'd be helpful to have a clearer explanation beforehand including what $k(s)$ represents.
>
> We will make the definition of $k(s)$ explicit where it first appears. In our algorithm, $k(s)$ denotes the index of the cluster to which sample $s$ is assigned in the current incumbent solution,
> $k(s) := \arg\min_{k \in \mathcal{K}} \|x_s - \mu^{\text{best}}_k\|_2^2.$
> We will add this clarification in Section 3.1 together with the other notation used in the geometric sample-determination rules.
>
> ---
>
> > **Q7**: You say $M_k$ represents any region, but then specify you mean an axis-aligned box. "Region" is a much more general term than "axis-aligned box" though, so I don't think you really mean "any" region.
>
> We appreciate the reviewer’s observation. **Our intention was not to claim full generality, but to model the search region for each centroid as an axis-aligned bounding box.** In practice, all geometric bounds and branch-and-bound nodes are implemented as such boxes. In the revised manuscript we will replace “any region” by “axis-aligned box (search region)” to avoid suggesting broader generality than what we actually use.
>
> > **Q8**: The notation for $M_k$ seems very close to $M_0$, which is used to mean something completely different later (the "solution space"). This is a bit confusing.
>
> In Subsection 3.1, we define $M_0 \subset \mathbb{R}^{mK}$ as the **initial axis-aligned box for the full centroid vector** $\mu = (\mu_1,\dots,\mu_K)$, whereas $M_k \subset \mathbb{R}^m$ denotes the **axis-aligned box for the individual centroid** $\mu_k$ used in the geometric rules. In other words, the per-cluster regions $M_k$ can be viewed as the projections of a branch-and-bound node $M \subseteq M_0$ onto the coordinates of centroid $k$.
>
> **For the camera-ready revision,** we will make this relationship explicit when $M_0$ is introduced and clarify that $M_k$ denotes the per-centroid projection of a node, to reduce any notational ambiguity.

---

> ### Author Response · Authors · 2025-11-17
> **Response to Reviewer 2GvY (3/3)**
>
> > **Q9**: In some places you write $d_{\min}(x_s, M_k)$, and in the figures you use $d_{\min}^i$ for some integer i. Can you give a more specific mapping between these notations so a reader can parse it more readily?
>
> In the main text, $d_{\min}(x_s, M_k)$ denotes the minimum possible squared distance between sample $x_s$ and any centroid in the region $M_k$. In the figures we fix a particular sample $x_s$ for illustration and simplify the notation by writing $d_{\min}^k := d_{\min}(x_s, M_k)$, so the superscript $k$ in $d_{\min}^k$ is exactly the cluster index $k$ in $d_{\min}(x_s, M_k)$. We hope this mapping makes the correspondence between the two notations clear.
>
> > **W2**: The technical contribution in 3.2 seems quite limited. In particular, this section seems to just be making the observation that if you include a must-link constraint, you can just collapse points into a new sample and then solving an equivalent unconstrained problem. This is a pretty basic and standard observation in constrained clustering, and it's significance is maybe a bit overstated.
>
> Collapsing must-link components is indeed not new at the modeling level. Our contribution in Section 3.2 is **not** to introduce representatives per se, but to **show that a specific ML pseudo-sample construction yields an unconstrained MSSC instance that is *exactly* equivalent to the original ML-only problem, and can be embedded into a deterministic RS–BB framework without losing global guarantees or scalability**.
>
> Concretely, in **Section 3.2 we prove** that collapsing each must-link component into repeated pseudosamples yields an unconstrained MSSC instance with identical global optimum to the original ML-only problem, and **Theorem 3.4 states that**
>
> >> *If $\mu^\*$ and $z(\mu^\*)$ are the global optimal solution and cost of Problem&nbsp;(4),  then they are also the global optimum and cost of the ML-only problem&nbsp;(2), and vice versa.*
>
> This result is precisely what allows us to plug ML pseudo-samples into the **reduced-space branch-and-bound (RS–BB)** scheme of Sections **3.3** and **3.4** while preserving all convergence and optimality guarantees: all lower bounds, pruning rules, and the global convergence proof in **Section 3.3 and Appendix C** continue to apply to the aggregated instance, so the $\varepsilon$–global optimality certificate is **preserved for the original ML-constrained MSSC**, not only for the pseudo-sample reformulation.
>
> While prior super–points similar approaches either
> 1. are exact but assignment/SDP-based and remain limited to small n (Piccialli et al.), or
> 2. are heuristic Lloyd-type methods without global bounds (Baumann, BLPKM-CC),
>
> to our knowledge prior works that employ representative/super-point constructions **do not combine this kind of centroid-space equivalence with a deterministic RS-BB algorithm and certified global guarantees at the scales we demonstrate in Section 4.1**.
>
> > **MC1** One final minor comment: the opening sentence in 3.3 is identical to the opening sentence in 3.4. Was this intended? It seems very repetitive.
>
> We thank the Reviewer for pointing out this detail, and we will remove the first sentence in Subsection 3.3 since it is indeed redundant.

---

> > ### Comment · Reviewer_2GvY · 2025-11-20
> > **Thanks for the response**
> >
> > Thanks for the careful and thoughtful responses. Your clarifications regarding my questions about the technical details in Section 3.1 are helpful.
> >
> > Thanks also for acknowledging that "collapsing must-link components is indeed not new at the modeling level", but then also expounding some more on what this technique accomplishes specifically in the setting you are considering. Collapsing must-link components to turn a constrained problem into an unconstrained problem is indeed a standard strategy in constrained optimization, so the novelty in the paper still feels a bit overstated to me in some places. That said, if a standard technique can be coupled with other advances leading to an algorithm that overall provides significant improvements in practice, that's still a good contribution, so I can appreciate what is being accomplished in this paper. And there is value in working out the details of this collapsing process in the context of a specific objective, even if the details are straightforward.
> >
> > In light of the response, I am increasing my score.

---

> > > ### Author Response · Authors · 2025-11-21
> > > **Official Comment by Authors**
> > >
> > > Thank you for the efforts spent on reviewing our paper and for the insightful questions. We are glad our response was useful!

---

### Official Review · Reviewer_9Ajy · 2025-10-30

**Soundness:** 3
**Presentation:** 3
**Contribution:** 3
**Rating:** 4
**Confidence:** 4

**Summary:**

The author introduces SDC-GBB, a novel branch-and-bound framework to solve the NP-hard Minimum Sum-of-Squares Clustering (MSSC) problem with constraints (e.g., must-link (ML) and cannot-link (CL)) that achieves global optimality on much larger instances. The algorithm branches only on the continuous centroid variables, avoiding the combinatorial explosion of assignment variables, and bounds are computed by a grouped-sample Lagrangian decomposition and a heuristic k-coloring approach.

The paper proposes several techniques to improve efficiency, such as collapsing each ML-connected component into repeated “pseudo-samples” at its centroid, and the geometric sample pruning rule to eliminate infeasible cluster assignment from geometric and constraint checks. Thus preserving the exact optimum while reducing problem size.

Empirically, SDC-GBB outperforms prior exact methods: it solves ML-only clustering on up to 1.5 million points and CL-only on up to 200,000 points.

**Strengths:**

- The pseudo-sample reformulation provably exact transformation with Lemma 3.3 and Theorem 3.4 shows that such a replacement shifts the objective by a constant, with the global minimum preserved.
- The method comes with a convergence proof that the BB scheme recovers the true optimum given exhaustive splitting.
- Geometric distance bounds to eliminate sample-to-cluster assignments, which prune the BB search tree aggressively before branching.
- SDC-GBB can handle huge datasets. The authors demonstrate orders-of-magnitude improvements in scale (up to 1.5M points) and achieve very low optimality gaps.

**Weaknesses:**

While the idea is interesting, some weaknesses in the proposed work's methodology need clarification:
- **Geometric Pruning Rules in High Cluster/High Dimensional Data**  Branch-and-bound on cluster centers lives in an $mK$-dimensional continuous space.  Geometric pruning rules are known to degrade in high dimensions or when K is relatively large;  they potentially weaken the elimination rules. The paper provides no theoretical justification or experimental analysis for $K>3$ or higher-dimensional data.  All experiments use low-dimensional data (2–8 features or 2D Gaussian mixtures) and a fixed small cluster count (K=3).
- **Computing Axis-bounding Box** For large feature spaces or larger $K$, $d_{\min}$ and $d_{\max}$ over each box $M_k$ may be loose if the region $M_k$ is large and can be costly to compute. The authors do not mention how the initial region $M_0$ is chosen or how boxes are subdivided in the methodology section.
- **High-variance ML components** The pseudo-sample approach relies on the covariance computations, which can be numerically unstable for large ML components. No theoretical or empirical analysis is given for more practical cases, such as a high-variance ML group.

Also, there are some things the authors can improve in the experiments:
- **Limited metric in experiments** The metric in the experiment only focuses on SSE. Since constraints often reflect domain needs, one might also evaluate clustering quality like the Rand Index. Also, the paper does not address whether the solutions actually respect all constraints.
- **Fraction of constraints** Also, the varying fraction of must-links vs. cannot-links in the dataset could dramatically affect difficulty, but no sensitivity analysis is presented.
- **Unclear statement** In the experiment design, some statement like “each dataset has $n/4$ samples bounded by ML/CL constraints” is unclear; it is not clear how many total constraint pairs this yields.
- **Reproducability** code and results are not provided in the current submission for reproducibility.

**Questions:**

-   How exactly are $d_{\min}(x_s, M_k)$ and $d_{\max}(x_s, M_k)$ computed for an axis-aligned box $M_k \subset \mathbb{R}^m$?
  Please clarify whether you use the closed-form per-coordinate formula implied in the text and how this scales with dimension.
-   For $\beta_{\text{SG+LD}}(M) := \max_{\lambda} \beta_{\text{SG+LD}}(M,\lambda)$, do you solve this dual at every BB node or only at nodes whose bounds are too loose?
- you use a single global $N = \max_{s,k} \sum_i \max\{|x_{s,i} - \mu^\ell_{k,i}|^2, |x_{s,i} - \mu^u_{k,i}|^2\}$, which can suffer from numerical or solver-sensitivity issues from large $N$. How did you resolve this?
- How key parameters (tolerance $\epsilon$, time limit, grouping sizes for Lagrangian decomposition) affect outcome and runtime?
- How does the algorithm perform with larger $K$ and higher-dimensional data?
- ablation study on different elements of the algorithms; for example, consider adding a baseline that first collapses ML components and then runs a standard $k$-means.
- it would be helpful for the authors to mention the trade-off between scalability and solution quality across methods. In particular, for the heuristic baselines, what are the corresponding wall-clock times and objective values as problem size increases, and how do these compare to the proposed approach?

---

> ### Author Response · Authors · 2025-11-18
> **Response to Reviewer 9Ajy (1/5)**
>
> We appreciate the opportunity to further clarify our contributions. **The core scope of our work is to advance the scalability of deterministic global optimization for constrained clustering**, enabling high-quality, reproducible solutions on datasets orders of magnitude larger than previously possible by exact methods.
>
> > **W1.** Branch-and-bound on cluster centers lives in an mK-dimensional continuous space. Geometric pruning rules are known to degrade in high dimensions or when K is relatively large; they potentially weaken the elimination rules. The paper provides no theoretical justification or experimental analysis for or higher-dimensional data. All experiments use low-dimensional data (2–8 features or 2D Gaussian mixtures) and a fixed small cluster count ($K=3$).
>
> **Our pruning strategy is not a generic geometric filter on the full $mK$-dimensional centroid space; it is a constraint propagation mechanism tailored to pairwise-constrained MSSC, and its correctness guarantees are independent of the ambient dimension $m$ and the number of clusters $K$.**
>
> In Section 3.1 (Geometric sample-determination rules, p. 4–5), Lemma 3.1 and Lemma 3.2 give sufficient conditions to eliminate or force specific sample–cluster assignments based on per-sample distance bounds:
>
> >> “for any sample $s \in S$ and cluster region $M_k$, if $d_{\min}(x_s, M_k) > \rho$ then $b_{s,k} = 0$ in every optimal solution with objective not larger than the incumbent, and if $d_{\max}(x_s, M_{k^+}) < \min_{k \neq k^+} d_{\min}(x_s, M_k)$ then $b_{s,k^+} = 1$ in every optimal solution.” (Section 3.1, p. 4–5)
>
> These tests act on the scalar quantities $d_{\min}(x_s, M_k)$ and $d_{\max}(x_s, M_k)$ for individual sample–cluster pairs and are then combined with propagation along the ML/CL graph. Figure 1 (p. 5) illustrates how geometry and pairwise constraints interact:
>
> >> “the cannot-link $(x_1, x_2)$ rules out $M^2$ for $x_2$, after which the geometric test fixes $x_2$ in $M^3$; in (d), the must-link $(x_2, x_3)$ propagates that assignment to $x_3$… If the ML/CL constraints forbid the move $(x_s \to k^+)$, the node becomes infeasible and is pruned.” (Figure 1, p. 5)
>
> Feasibility with respect to cannot-links is enforced by the $K$-coloring bound in Section 3.3 (p. 6), which tests the CL graph directly:
>
> >> “the labeling $\chi^{(c)}$ is proper if $\chi^{(c)}(u) \neq \chi^{(c)}(v)$ for every $(u,v) \in T_{cl}$; otherwise we set $z^{(c)}_{\text{ub}} = +\infty$.” (Section 3.3, p. 6)
>
> This guarantees that every upper bound used for branching is compatible with all cannot-links, and the coloring bound itself does not depend on the dimension of the feature space.
>
> **That means that the mechanism that enforces feasibility and propagates ML/CL constraints is dimension agnostic and valid for any $m$ and any $K$.** The geometric tests in Section 3.1 serve to identify assignments already determined by the current centroid regions and to trigger propagation in the ML/CL graph. If the ambient dimension or the number of clusters increases, the algebraic conditions in Lemma 3.1 and Lemma 3.2 remain valid and the coloring bound continues to enforce feasibility; only the frequency with which these inequalities fire may change, which affects efficiency but not the correctness of the propagated assignments or the guarantees provided by the coloring construction.
>
> ---
>
> > **W2.1**: For large feature spaces or larger $K$, $d_{\min}$ and $d_{\max}$ over each box $M_k$ may be loose if the region $M_k$ is large and can be costly to compute.
>
> **In our framework, $M_k$ is an axis-aligned box in $\mathbb{R}^m$, and $d_{\min}(x_s,M_k)$, $d_{\max}(x_s,M_k)$ are the exact minimum and maximum squared Euclidean distances over that box, not relaxations.** Section 3.1 (p. 4–5) states:
>
> >> “for any region $M_k$ (an axis–aligned box in $\mathbb{R}^m$) containing the true optimal $\mu_k$, we can compute the minimal and maximal possible squared distances $d_{\min}(x_s, M_k) = \min_{\mu \in M_k} \lVert x_s - \mu \rVert_2^2$
> and
> $d_{\max}(x_s, M_k) = \max_{\mu \in M_k} || x_s - \mu ||_2^2$.”  (Section 3.1, p. 4–5)
>
> Since the box is axis-aligned and the distance is separable by coordinate, these quantities are obtained in closed form by clamping each coordinate of $\mu$ to the bounds of $M_k$, which is an $O(m)$ computation per $(s,k)$ pair. Algorithm 1 (Appendix A, p. 14) shows that this geometric sample-determination step is applied as a preprocessing stage before branching, rather than as a heavy inner optimization at every node.
>
> **Thus, computing $d_{\min}$ and $d_{\max}$ is inexpensive even in large feature spaces, and any “looseness” for wide boxes is systematically removed by the axis-aligned subdivision of $M_k$ in branch-and-bound, whose convergence is guaranteed by Theorem 3.5 (Section 3.5 and Appendix C, p. 7 and p. 16).**

---

> ### Author Response · Authors · 2025-11-18
> **Response to Reviewer 9Ajy (2/5)**
>
> > **W2.2**: The authors do not mention how the initial region $M_0$ is chosen or how boxes are subdivided in the methodology section.
>
> **In our implementation, the initial region $M_0$ is obtained by building an axis-aligned bounding box for all centroids in feature space.** This region is defined in **Appendix A (Two-stage programs reformulation, p. 13–14)**:
>
> > > “The closed set $M_0 = {\mu \mid \mu^l \le \mu \le \mu^u}$ bounds every centre with $\mu_{k,i}^l = \min_s X_{s,i}$ and $\mu_{k,i}^u = \max_s X_{s,i}$ for all $k \in K$ and $i = 1,\dots,m.$”
>
> **Subdivision follows the standard exhaustive axis-aligned bisection framework, adopted from Cao & Zavala (2019) and made explicit in Section 3.5 and Algorithm 1.**
>
> > > “We adopt the framework of the reduced-space branch-and-bound scheme from (Cao & Zavala, 2019) and tailor the algorithm for the pairwise constrained clustering task.”
>
> At each branching step, a node $M \subseteq M_0$ is split into two child boxes by bisecting one coordinate interval of one centroid (for example, along the dimension with the largest extent). This produces progressively smaller $M_k$ and therefore tighter $d_{\min}(x_s,M_k)$ and $d_{\max}(x_s,M_k)$ as the search proceeds.
>
> ---
>
> > **W3.** The pseudo-sample approach relies on the covariance computations, which can be numerically unstable for large ML components. No theoretical or empirical analysis is given for more practical cases, such as a high-variance ML group.
>
> **We respectfully disagree and clarify that the pseudo-sample construction uses only the trace of the covariance, which is a simple sum of squared deviations.** Specifically, in **Section 3.2 (p. 4)** we define
>
> > > “$\mathrm{tr}(\Sigma_{\mathrm{ml}}^2) = \frac{1}{t-1}\sum_{i=1}^t |x_i - \mu_{\mathrm{ml}}|^2$.”
>
> Using **Lemma 3.3 (p. 4)**, we show that for each must-link component,
>
> >> "$sse_{\mathcal{C}}(\mu) = sse_{\hat{\mathcal{C}}}(\mu) + (t-1)tr(\Sigma_{ml}^2).$"
>
> As made explicit in **Appendix A (p. 14)**,
>
> > > “Once the must-link set $\mathcal T_{ml}$ has been collapsed into pseudo-samples (Sec. 3.2), the additive constant $(t-1),\mathrm{tr}(\Sigma_{\mathrm{ml}}^2)$ can be pre-computed. Hence minimising the objective in (4a) is equivalent to minimising $\sum_{s\in\hat{\mathcal S}} d_{s,*}$.”
>
> **Thus $(t_k-1)\mathrm{tr}(\Sigma_{\mathrm{ml},k}^2)$ is computed once per ML component as an additive constant and then dropped from the optimization over centroids.** Any floating-point rounding in this term affects only the absolute SSE value, not the location of the minimizer, and therefore does not impact the convergence or correctness of SDC-GBB. **In the camera-ready version, we will make this explicit by adding: “Note that $(t_k-1)\mathrm{tr}(\Sigma_{\mathrm{ml},k}^2)$ is an additive constant that does not affect the optimization over centroids.”**
>
> ---
>
> > **W4** The metric in the experiment only focuses on SSE. Since constraints often reflect domain needs, one might also evaluate clustering quality like the Rand Index.
>
> Our experimental design follows a **standard distinction in the clustering literature** ([1,2]), which we explicitly state in **Appendix D.2 (Clustering Evaluation)**:
>
> >> Our work distinguishes between **two aspects of clustering performance**: (i) the formulation, which defines how data similarity is measured and affects metrics such as ARI, NMI, and purity; and (ii) the quality of the solution (in terms of cost minimization), which we measure using the optimality gap. (App. D.2)
>
> **Given that the primary contribution of our work is a deterministic global optimizer for the MSSC task under hard pairwise constraints**, and that we **provide optimality guarantee at an unprecedented scale** compared to previous exact methods, **SSE and the optimality gaps are natural solution quality metrics as shown in the clustering literature [3,4,5,6]**.
>
> Nonetheless, we still report on other clustering quality metrics, namely ARI, NMI and Purity on labelled real-world datasets in **Table 6, p.20**. We show that across all constraint types and datasets, SDC-GBB matches or performs better than the baseline in Hua et al. (2021), suggesting that our globally optimal solutions correspond to meaningful clusters.
>
> * [1] Ben-David & Ackerman, “Measures of Clustering Quality: A Working Set of Axioms for Clustering”, NeurIPS 2008.
> * [2] Jain, “Data Clustering: 50 Years Beyond k-Means”, Pattern Recognition Letters 2010.
> * [3] Hua et al., “A Scalable Deterministic Global Optimization Algorithm for Clustering Problems”, ICML 2021.
> * [4] Piccialli, V. et al. (2022). An exact algorithm for semi-supervised minimum sum-of-squares clustering. Computers & Operations Research.
> * [5] Babaki, B. et al. (2014). Constrained clustering using column generation. CPAIOR.
> * [6] Guns, T. et al. (2016). Repetitive branch-and-bound using constraint programming for constrained minimum sum-of-squares clustering. ECAI.

---

> ### Author Response · Authors · 2025-11-19
> **Response to Reviewer 9Ajy (3/5)**
>
> > **W5.** Also, the paper does not address whether the solutions actually respect all constraints.
>
> **We respectfully disagree, as we have stated that constraint satisfaction is guaranteed at every stage of the algorithm, both theoretically and computationally.**
>
> **Section 3.1 (Page 3)** explicitly states that any constraint violation triggers pruning:
> >> "If the ML/CL constraints forbid the move $(x_s \to k^+)$, **the node becomes infeasible and is pruned**."
>
> The geometric rules in **Lemmas 3.1-3.2 (Page 3)** eliminate assignments that would violate constraints *before* branching, as shown in **Figure 1 (c)-(d)** where:
> >> "*cannot-link: $b_{s,2}=0 \Rightarrow$ L3.2 fixes $b_{s,3}=1$ and must-link: link propagation fixes $b_{s,3}=1$.*"
>
> **Section 3.3 (Page 5)** enforces constraints in upper bounding:
> >> "The labeling $\chi^{(c)}$ is **proper if $\chi^{(c)}(u) \neq \chi^{(c)}(v)$ for every $(u,v) \in \mathcal{T}_{cl}$**. Define $z_{ub}^{(c)} = ...$ **If $\chi^{(c)}$ is proper, then $z_{ub}^{(c)}$ satisfies all CL constraints** and $z(M) \le \alpha(M)$."
>
> For ML constraints, **Theorem 3.4 (Page 5)** proves global equivalence after collapsing must-link components, ensuring the transformed problem preserves all original constraints:
> >> "If $\mu^*$ and $z(\mu^*)$ are the global optimal solution and cost of Problem (4), **then they are also the global optimum and cost of the ML-only problem (2), and vice versa**."
>
> **Algorithm 1 (Page 7)** includes constraint propagation:
> >> "Propagate fixes via $\mathcal T_{ml}$, $\mathcal{T}_{cl}$; update $\\{K_s\\}$;
>
> These mechanisms ensure that **any solution returned by SDC-GBB necessarily satisfies all pairwise constraints**. All violations are pruned during search and prohibited in bound evaluation.
>
> > **W6.** The varying fraction of must-links vs. cannot-links in the dataset could dramatically affect difficulty, but no sensitivity analysis is presented.
>
> **In **Appendix D (Page 17-19)**, we provide a comprehensive sensitivity analysis in Table 4 that systematically evaluates constraint densities ranging from $n/64$ to $n/2$, covering very sparse to very dense regimes**.
>
> The table explicitly reports:
> * Branch-and-bound nodes processed
> * Wall-clock time and per-node time
> * Core-hour consumption
> * Optimality gap (all experiments achieve $\le 0.1\%$)
>
> Appendix D further analyzes how ML constraints reduce node count but increase per-node complexity, while CL constraints scale node count linearly with density, while maintaining solution quality. This directly validates robustness across constraint fractions.
>
> > **W7.** In the experiment design, some statement like “each dataset has $n/4$ samples bounded by ML/CL constraints” is unclear; it is not clear how many total constraint pairs this yields.
>
> We would like to clarify that we generate $n/4$ pairs of constraints for the ML-only and CL-only cases, and combine these for the "both constraints" case, resulting in $n/2$ pairwise constraints.
>
> > **W8.** Reproducibility code and results are not provided in the current submission for reproducibility.
>
> To support the reproducibility of our experiments, we have included implementation steps and computational settings in **Section 4 (Page 6-7) and Appendix D**. *We will release the code, datasets, and constraints during the camera-ready submission*.

---

> ### Author Response · Authors · 2025-11-21
> **Response to Reviewer 9Ajy (4/5)**
>
> > **Q1.** How exactly are $d_{\min}(x_s, M_k)$ and $d_{\max}(x_s, M_k)$ computed for an axis-aligned box $M_k \subset \mathbb{R}^m$? Please clarify whether you use the closed-form per-coordinate formula implied in the text and how this scales with dimension.
>
> In Section 3.1 (Geometric sample-determination rules, pages 3–4) we define $M_k$ as an axis-aligned box and $d_{\min}$, $d_{\max}$ as the exact minimum and maximum squared Euclidean distances from $x_s$ to that box:
>
> >> “for any region $M_k$ (an axis–aligned box in $\mathbb{R}^m$) containing the true optimal $\mu_k$, we can compute the minimal and maximal possible squared distances  $d_{\min}(x_s, M_k) = \min_{\mu \in M_k} \lVert x_s - \mu \rVert_2^2$, and $d_{\max}(x_s, M_k) = \max_{\mu \in M_k} \(\lVert x_s - \mu \rVert_2^2\).$
>
> Concretely, if $M_k = \prod_{i=1}^m [\ell_{k,i}, u_{k,i}]$ and $x_s = (x_{s,1}, \dots, x_{s,m})$, we use the closed-form per-coordinate formulas implied by this definition. For each coordinate we set
>
>
>
> $ \Delta_{s,k,i}^{min} = 0 \text{ if } \ell_{k,i} \le x_{s,i} \le u_{k,i}.$
>
> $ \Delta_{s,k,i}^{min} = \ell_{k,i} - x_{s,i}$ if $x_{s,i} < \ell_{k,i}$,
>
> $ \Delta_{s,k,i}^{min} = x_{s,i} - u_{k,i}$ if $x_{s,i} > u_{k,i}$,
>
> and
>
> $ \Delta_{s,k,i}^{min} = \max\(|x_{s,i} - \ell_{k,i}|,\ |x_{s,i} - u_{k,i}|\)$.
>
> The distance bounds then follow as
> $d_{\min}(x_s, M_k) = \sum_{i=1}^m (\Delta_{s,k,i}^{\min})^2$ and $d_{\max}(x_s, M_k) = \sum_{i=1}^m (\Delta_{s,k,i}^{\max})^2$.
>
> Geometrically, $d_{\min}$ corresponds to projecting $x_s$ onto the box, and $d_{\max}$ corresponds to the distance to the farthest vertex of the box. No inner optimization problem is solved, and the cost of computing both $d_{\min}$ and $d_{\max}$ for a given pair $(s,k)$ is $O(m)$.
>
> The same per-coordinate structure appears in Appendix A (Two-stage programs reformulation, page 14), where we define the global big-$M$ constant as
>
>
> $N = \max_{s,k} \sum_{i=1}^m max\(\lvert x_{s,i} - \mu_{k,i}^{\ell} \rvert^2, \lvert x_{s,i} - \mu_{k,i}^{u}\rvert^2\)$,
>
>
>
> which is exactly the coordinate-wise extremal squared distance from $x_s$ to the bounding box $[\mu_{k,i}^{\ell}, \mu_{k,i}^{u}]$. Our implementation therefore already uses the same closed-form per-coordinate formulas. The overall cost of computing all $\{d_{\min}, d_{\max}\}_{s,k}$ at the root node is $O(|\hat S| K m)$. We will make these formulas and their $O(m)$ scaling explicit in Section 3.1 in the revised manuscript.
>
>
> ---
>
> > **Q2.** For $\beta_{\mathrm{SG+LD}}(M) := \max_{\lambda} \beta_{\mathrm{SG+LD}}(M,\lambda)$, do you solve this dual at every BB node or only at nodes whose bounds are too loose?
>
> We solve the grouped Lagrangian dual $\beta_{\mathrm{SG+LD}}(M)$ at every branch-and-bound node where a lower bound is required. This is the lower-bounding function $\beta$ used in Algorithm 1, not a fallback used only at difficult nodes.
>
> Section 3.4 (Lower bounding strategy with grouping-based Lagrangian decomposition, page 6) explains that the branch-and-bound algorithm “requires a fast yet tight lower bound for each subproblem or node $M$ inside the solution space $M_0$” and that dualizing the coupling constraints with multipliers $\lambda$ “yields a tighter lower bound via $\beta_{\mathrm{SG+LD}}(M) := \max_{\lambda} \beta_{\mathrm{SG+LD}}(M,\lambda)$.”
>
> Algorithm 1 (Branch-and-Bound Clustering with Geometric Sample Determination, page 6) then uses this $\beta$ systematically at the root and after each branching step:
>
> >> “Compute upper bound $\alpha(M_0)$, lower bound $\beta(M_0)$” and, after splitting a node $M$, “Compute $\alpha(M_1), \beta(M_1), \alpha(M_2), \beta(M_2)$.”
>
> In our implementation, $\beta$ in Algorithm 1 is instantiated as $\beta_{\mathrm{SG+LD}}$ for all nodes. The sample grouping is fixed once at the root node (“The grouping is fixed at the root for efficiency”, Section 3.4, page 6), so the grouped subproblem structure is shared across all nodes and the dual problem (6) is solved consistently at every node used in the search.

---

> ### Author Response · Authors · 2025-11-21
> **Response to Reviewer 9Ajy (5/5)**
>
> > **Q3.** You use a single global
> >
> > $N = \max_{s,k} \sum_i \max\(|x_{s,i} - \mu_{k,i}^{\ell}|^2,\ |x_{s,i} - \mu_{k,i}^{u}|^2\)$,
> >
> > which can suffer from numerical or solver-sensitivity issues from large $N$. How did you resolve this?
>
> The big-$M$ constant $N$ is the minimal valid uniform bound implied by the data and the centroid domain $M_0$.
>
> In Appendix A (Two-stage programs reformulation, page 14) we define $M_0 = \{\mu \mid \mu^{\ell} \le \mu \le \mu^{u}\}$, where:
>
> * $\mu_{k,i}^{\ell} = \min_s X_{s,i}$,
> * $\mu_{k,i}^{u} = \max_s X_{s,i}$ for all $k,i$,
>
> and set:
>
> $N := \max_{s,k} \sum_{i=1}^m \max\{|x_{s,i} - \mu_{k,i}^{l}|^2,\ |x_{s,i} - \mu_{k,i}^{u}|^2\}.$
>
> For any $\mu \in M_0$ and any $(s,k)$ we have:
>
> $d_{s,k} = \lvert x_s - \mu_k\rvert_2^2 \le \sum_{i=1}^m \max\(\lvert x_{s,i} - \mu_{k,i}^{l}\rvert^2, \lvert x_{s,i} - \mu_{k,i}^{u}\rvert^2\) \le N,$
>
> so, since $d_{s,*} \ge 0$, the linking constraint
>
> $-N(1 - b_{s,k}) \le d_{s,*} - d_{s,k} \le N(1 - b_{s,k})$
>
> is valid for every feasible $(d_{s,*}, d_{s,k})$ whenever $\mu \in M_0$.
>
> Any smaller uniform constant would violate this inequality for some $(s,k,\mu)$, so this $N$ is the smallest global big-$M$ that keeps (2b) correct over $M_0$. The bounds $\mu^{\ell}$, $\mu^{u}$ and $N$ are computed once at the root from the data extremes and inherited unchanged by all subproblems.
>
> > **Q4.** It would be helpful for the authors to mention the trade-off between scalability and solution quality across methods. In particular, for the heuristic baselines, what are the corresponding wall-clock times and objective values as problem size increases, and how do these compare to the proposed approach?
>
> Section 4 (Computational Experiments, p. 7) specifies that all heuristic baselines use a fixed computational budget of 100 restarts within 4 hours, with full results detailed in Appendix E. **Consequently, we evaluate the trade-off by comparing the solution quality achieved within this time limit against the certified bounds of our method**.
>
> Tables 1–3 (pp. 8–9) report the best objective values obtained by each algorithm. **These results demonstrate that SDC-GBB matches or exceeds heuristic solution quality while certifying optimality gaps $\le 0.1\%$ on small and medium datasets, and maintains gaps $\le 3\%$ on large instances where heuristics often fail to find feasible solutions.**
>
> **Regarding the runtime scaling of the proposed method**, Appendix D.1 (p. 18) provides detailed wall-clock statistics (Table 5) as the sample size increases from 21,000 to 171,000. For the camera-ready revision, we will include a statement in Section 4 synthesizing this comparison to highlight that SDC-GBB delivers superior cost-quality guarantees under comparable time constraints.
>
> > **Q5.** How does the algorithm perform with larger $K$ and higher-dimensional data?
>
> The validity of our geometric pruning rules is independent of the number of clusters $K$ or the feature dimension $m$; they remain correct for arbitrary $K$ and $m$. Computationally, the per-node cost scales linearly with the feature dimension ($O(m)$) because distance bounds are computed coordinate-wise.
>
> As with any exact branch-and-bound scheme, the total number of nodes grows as $K$ and $m$ increase. However, in the constrained setting, the propagation of ML/CL constraints (via pseudo-sample collapse and elimination) reduces the search tree. In our experiments, which cover moderate values of $K$ and low to medium feature dimensions, the method remains tractable due to these pruning mechanisms.

---

> ### Author Response · Authors · 2025-11-23
> **Follow-up – Does This Address Your Concerns?**
>
> Dear Reviewer 9Ajy,
>
> As the discussion period closes, we summarize our response regarding **geometric pruning, stability, and methodology**.
>
> We clarified that our geometric rules remain theoretically valid for any dimension $m$ and cluster count $K$ using exact $O(m)$ bounds. We also confirmed that pseudo-sample collapse ensures numerical stability via pre-computed constants, and that constraint satisfaction is strictly enforced at every node.
>
> *Do these clarifications resolve your doubts regarding the contribution and technical novelty? If not, please let us know so we can address any remaining points.*
>
> Best regards,
>
> Authors of Paper 14782

---

### Official Review · Reviewer_Jhaj · 2025-10-30

**Soundness:** 3
**Presentation:** 3
**Contribution:** 2
**Rating:** 4
**Confidence:** 5

**Summary:**

The paper introduces SDC-GBB, a deterministic global optimization framework for constrained clustering. It combines geometric pruning, pseudo-sample aggregation, and Lagrangian decomposition to scale branch-and-bound methods to much larger datasets. The work is clearly written and technically careful.

**Strengths:**

Addresses a hard and relevant problem. The algorithmic framework is well motivated and mathematically sound. Most parts of experiments are extensive and the implementation seems professional. The writing quality is high overall.

**Weaknesses:**

The conceptual novelty is limited — most components (B&B structure, geometric elimination, decomposition) have appeared in previous global optimization literature. The claimed scalability relies heavily on large-scale hardware rather than clear algorithmic gains. Experimental comparisons are somewhat narrow, mostly against older baselines. The contribution feels more like a well-engineered variant than a new learning idea.

**Questions:**

1. Could the authors clarify what parts of SDC-GBB are genuinely novel compared to Cao & Zavala (2019) and Piccialli et al. (2022)? 2. How much of the reported speedup arises from parallelization versus algorithmic improvement? 3. What is the computational complexity per iteration or per branch node? Any empirical scaling curve (e.g., runtime vs n)? 4. Would the method still converge on moderate hardware (e.g., 32 cores, 64 GB RAM)? 5. Can the same framework be extended to non-Euclidean or kernelized distances?

I would be glad to improve my scores if the authors could resolve my doubts.

---

> ### Author Response · Authors · 2025-11-18
> **Response to Reviewer Jhaj (1/4)**
>
> We appreciate the opportunity to further clarify our contributions. **The core scope of our work is to advance the scalability of deterministic global optimization for constraint clustering**, enabling high-quality, reproducible solutions on datasets orders of magnitude larger than previously possible by exact methods.
>
> > **W1.** The conceptual novelty is limited — most components (B&B structure, geometric elimination, decomposition) have appeared in previous global optimization literature.
>
> **Existing global optimization methods for MSSC cannot be applied directly once hard ML/CL constraints are introduced, and none of them decomposes these constraints while retaining exact global guarantees and scalability.**
>
> In the unconstrained MSSC, the assignment stage decomposes across samples: given the centroids, each point can be treated independently. Hard must-link and cannot-link constraints ($b_{s,k} = b_{s',k}$ and $b_{s,k} + b_{s',k} \le 1$) couple these assignments across samples, so the second stage no longer decomposes. Simply appending ML/CL constraints to an existing global $k$-means solver breaks both the lower-bounding mechanism and the feasibility of the upper bounds; global optimality guarantees are lost for the constrained MSSC.
>
> Our contribution is precisely to make the pairwise-constrained MSSC globally solvable again at scale, by recovering decomposability and correctness **exactly**:
>
> 1. Geometric sample determination with constraint propagation (Sec. 3.1). New geometric rules combine distance bounds with ML/CL propagation to fix or eliminate assignments while preserving feasibility at every node.
>
> 2. ML collapse equivalence (Sec. 3.2, Thm. 3.4). We prove that collapsing each must-link component into centroid-based pseudo-samples yields an **unconstrained** MSSC instance with the **same global minimizer and objective value** as the original ML-only problem. This is an exact equivalence, not a heuristic, and it is what allows us to run a reduced-space search on the pseudo-sample instance without losing global optimality for the original constrained problem.
>
> 3. Constraint-aware bounds (Sec. 3.3–3.4). We design a \(k\)-coloring-based upper bound for CL constraints and a grouped Lagrangian lower bound on the pseudo-sample instance. These bounds explicitly incorporate the pairwise constraints and yield valid global lower bounds for the original constrained MSSC.
>
> On top of these theoretical and algorithmic ingredients, we provide the **first deterministic $\varepsilon$-global guarantees for MSSC with hard pairwise constraints at large scale**, solving ML-constrained instances up to **1.5M** samples and CL instances up to **200k** samples—roughly **7× larger** than previous global MSSC experiments and about **200–1500× beyond existing exact constrained methods.**
>
> ---
>
> > **W2.** The claimed scalability relies heavily on large-scale hardware rather than clear algorithmic gains.
>
> We respectfully disagree with this assessment. The dramatic increase in scalability ($200$–$1500\times$ over prior exact methods) stems from algorithmic innovations that **restore a fully decomposable structure at each branch-and-bound node, enabling essentially embarrassingly parallel lower-bound computations.** Specifically:
>
> - The geometric sample-determination rules (Lemmas 3.1–3.2) and must-link pseudo-sample construction (Theorem 3.4) eliminate most of the coupling introduced by pairwise constraints.
> - Grouped-sample Lagrangian decomposition (Section 3.4) and centroid-only branching then yield independent subproblems per group that require virtually no inter-process communication.
>
> **This decomposable structure is a direct consequence of our algorithmic design and is not present in previous exact methods for constrained MSSC** (including MISOCP formulations, column-generation, constraint-programming, or PC-SOS-SDP approaches), which retain tightly coupled assignment variables. **Parallel hardware merely distributes these independent, much cheaper node relaxations across cores**; it does not tighten the bounds, reduce the search tree, or affect the provable $\varepsilon$-global optimality guarantees. Thus, the scalability gain is primarily algorithmic, with large-scale hardware serving only to realise the inherent parallelism that our method creates.

---

> ### Author Response · Authors · 2025-11-18
> **Response to Reviewer Jhaj (2/4)**
>
> > **W3.** Experimental comparisons are somewhat narrow, mostly against older baselines.
>
> **This is not accurate: our comparisons include modern heuristics as well as the strongest available exact solvers for constrained MSSC.**
>
> We evaluate against state-of-the-art heuristics, such as encode-kmeans-post (Nghiem, 2020) and BLPKM-CC (Baumann, 2020), together with the classical COP-k-means baseline (Wagstaff et al., 2001). On the exact side, we test against the exact CPLEX solver and **the leading exact algorithm PC-SOS-SDP (Piccialli, 2022)**.
>
> **No exact method before or after Piccialli et al. (2022) is able to handle constrained MSSC beyond a few thousand samples**, making it the only viable exact benchmark at this scale. As shown in Tables 1–3, SDC-GBB consistently attains ≤3% optimality gaps on instances where exact baselines either time out or fail to find any feasible solution, and matches or exceeds the best heuristic SSE while providing deterministic global guarantees.
>
> ---
>
> > **W4.** The contribution feels more like a well-engineered variant than a new learning idea.
>
> **We respectfully disagree: in the context of semi-supervised clustering with pairwise constraints, our contribution changes what is computationally achievable for a standard learning objective.**
>
> From the learning side, **pairwise-constrained MSSC is a canonical semi-supervised clustering problem** [8–10]. It is NP-hard even to decide feasibility or find a good approximate solution [2,3], and surveys consistently describe constrained clustering as dominated by heuristics without global guarantees [10,11]. On the optimization side, deterministic global methods exist only for the *unconstrained* MSSC and remain limited to medium/big-scale datasets (200k samples) [1,4]. For the **constrained** MSSC, all exact approaches to date (column generation, CP, MIP/SDP formulations) scale only to hundreds or a few thousand points in practice [5–7].
>
> **Our contribution is to close this gap between global optimization and semi-supervised learning:**
>
> - We provide the **first deterministic ε-global optimizer for the exact pairwise-constrained MSSC objective at truly large scale**, solving ML-constrained instances up to 1.5M points and CL instances up to 200k points, where previous exact methods stop at ≪10⁴ [5–7].
> - We do this **without changing the learning objective**: we optimize the *same* MSSC loss studied in k-means hardness and approximation work [2], and the same pairwise-constrained objective used in semi-supervised clustering and side-information literature [8–10].
> - Algorithmically, we extend deterministic **global** methods from the unconstrained setting [1,4] to the full ML/CL case while preserving exactness and scalability, something no prior constrained method achieves [5–7].
>
> In this sense, the work is not just “engineering”: it **revises the practical boundary of what the learning community can do with a classical semi-supervised clustering model**, by making its exact global optimum accessible at scales previously handled only by heuristics.
>
> [1] Horst, R., & Tuy, H. (2013). *Global Optimization: Deterministic Approaches*. Springer.
> [2] Dasgupta, S. (2008). *The hardness of k-means clustering*.
> [3] Davidson, I., & Ravi, S. S. (2007). *Intractability and clustering with constraints*. ICML.
> [4] Hua, K. et al. (2021). *A scalable deterministic global optimization algorithm for clustering problems*. ICML.
> [5] Piccialli, V. et al. (2022). *An exact algorithm for semi-supervised minimum sum-of-squares clustering*. *Computers & Operations Research*.
> [6] Babaki, B. et al. (2014). *Constrained clustering using column generation*. CPAIOR.
> [7] Guns, T. et al. (2016). *Repetitive branch-and-bound using constraint programming for constrained minimum sum-of-squares clustering*. ECAI.
> [8] Wagstaff, K. et al. (2001). *Constrained k-means clustering with background knowledge*. ICML.
> [9] Basu, S., Bilenko, M., & Mooney, R. (2004). *A probabilistic framework for semi-supervised clustering*. KDD.
> [10] Basu, S., Davidson, I., & Wagstaff, K. (2008). *Constrained Clustering: Advances in Algorithms, Theory, and Applications*. Chapman & Hall/CRC.
> [11] Davidson, I., & Basu, S. (2007). *A survey of clustering with instance-level constraints*. *ACM TKDD*.

---

> ### Author Response · Authors · 2025-11-18
> **Response to Reviewer Jhaj (3/4)**
>
> > **Q1**. Could the authors clarify what parts of SDC-GBB are genuinely novel compared to Cao & Zavala (2019) and Piccialli et al. (2022)?
>
> > **Q1.1** Regarding Cao & Zavala (2019)
>
> Cao & Zavala (2019) study a two-stage stochastic NLP with continuous recourse. After relaxing non-anticipativity, the recourse subproblems $Q_s(x)$ become independent across scenarios, and their reduced-space branch-and-bound branches only on the first-stage continuous variables, using a “wait-and-see’’ lower bound, which can be interpreted as a Lagrangian relaxation with dual multipliers set to zero.
>
> In **SDC-GBB**, the recourse stage is a **mixed-integer assignment problem with hard ML/CL constraints**, modeled as a MISOCP. These constraints couple samples, **so the scenario-wise separability obtained in Cao & Zavala (2019) does not apply**.
>
> To recover a reduced-space scheme in this setting, SDC-GBB introduces three structures that **do not appear in Cao & Zavala (2019):**
>
> - an *exact pseudo-sample collapse theorem for ML constraints*, mapping the ML-only problem to an unconstrained MSSC with the same global optimum in centroid space,
> - *geometric sample-determination rules* that prune or fix binary assignments while preserving all ML/CL constraints, and
> - a *grouped Lagrangian lower bound* that enforces non-anticipativity within pseudo-sample groups and relaxes it only across groups, strengthening the basic wait-and-see (dual = 0) relaxation.
>
> These structures are specific to pairwise-constrained MSSC with mixed-integer recourse and do not appear in the continuous reduced-space BB framework of Cao & Zavala (2019).
>
>
> > **Q1.2** Regarding Piccialli et al. (2022)
>
> In **PC-SOS-SDP (Piccialli et al., 2022)**, the branch-and-cut lower bound at each node is an SDP in the **$n \times n$** cluster matrix $Z$. This implies that
>
> - the decision space at each node has $O(n^2)$ parameters (all entries of $Z$), and
> - solving the SDP relaxation at one node requires at least cubic time in $n$ in practice (standard interior-point methods for SDPs).
>
> In **SDC-GBB (our work)**, branch-and-bound never branches in the $n \times n$ assignment space. Instead:
>
> - we branch only on the **centroids** $\mu \in \mathbb{R}^{dK}$, so the **search space has dimension $dK$, independent of $n$**,
> - assignments and ML/CL constraints are handled in recourse (via pseudo-samples, geometric rules, and grouped Lagrangian bounds), which leads to **per-node work that grows essentially linearly with $n$**.
>
> **In short, Piccialli’s method explores an $n^2$-dimensional matrix space at each node, while SDC-GBB explores only a $dK$-dimensional centroid space; this structural difference is exactly what allows us to scale to much larger $n$.**
>
> ---
>
> > **Q2.** How much of the reported speedup arises from parallelization versus algorithmic improvement?
>
> **Most of the scalability gain comes from the algorithmic structure of SDC-GBB; parallelization mainly yields an almost linear reduction in wall-clock time on top of that.**
>
> In our complexity analysis **(Section D, *Effect of Pairwise Constraints*, p. 17-19)**, we show that each branch-and-bound node and the total number of nodes satisfy:
> >> "each branch-and-bound node incurs $O(|\hat S|K)$ work, while exhaustive axis-aligned bisection yields at most $O((\delta/\varepsilon)^K m)$… Together these give a nominal worst-case cost of $O((\delta/\varepsilon)^K m |\hat S| K)$. On a cluster with $P$ identical CPU cores, parallelising across open nodes yields an expected core-hour consumption of $O((\delta/\varepsilon)^K m |\hat S| K / P)$, provided the number of simultaneously available nodes exceeds $P$."
>
> **Thus, in serial the worst-case cost is $O((\delta/\varepsilon)^K m |\hat S| K)$, and using $P$ cores simply distributes independent open nodes, giving wall-clock scaling of order $O((\delta/\varepsilon)^K m |\hat S| K / P)$.**
>
> To separate algorithmic effects from parallelism, we define and report **core-hours** in the same section:
> >> "the wall-clock time increases by about 20% on Syn-1,200 and roughly a factor of two on Syn-2,100, but the core hour stays relatively static across constraint sizes."
>
> Empirically, when we vary the constraint density, the *number of nodes* and *time per node* change, but core-hours remain nearly constant. **This shows that the ability to solve constrained MSSC at the reported scales is driven primarily by reducing node count and per-node cost (pseudo-sample collapse, geometric ML/CL pruning, grouped Lagrangian bounds); large-scale hardware only speeds up an already more efficient global algorithm**.

---

> ### Author Response · Authors · 2025-11-18
> **Response to Reviewer Jhaj (4/4)**
>
> > **Q3**. What is the computational complexity per iteration or per branch node? Any empirical scaling curve (e.g., runtime vs n)?
>
> The global complexity result quoted in Q2 follows from a more basic quantity: the cost per branch-and-bound node. In Appendix D (p. 18), we establish that
> >> "each branch-and-bound node incurs $O(|\hat S|K)$ work," (Appendix D, p. 19)
>
> which is the fundamental per-node complexity of SDC-GBB. The total number of nodes follows from the geometry of the exhaustive axis-aligned subdivision of the centroid space, **giving (as cited in Q2) a worst-case envelope of $O((\delta/\varepsilon)^K m)$**.
>
>
> **Thus, in serial, the worst-case cost per node is $O(|\hat S|K)$, the worst-case number of nodes is $O((\delta/\varepsilon)^{K m})$, and the resulting nominal worst-case serial cost is
> $O((\delta/\varepsilon)^{K m} |\hat S| K)$.**
>
> For empirical scaling with respect to $n$, **Appendix D.1 (p. 19)** reports detailed runtime statistics for synthetic datasets ranging from $21{,}000$ to $171{,}000$ samples (Table 5). There we decompose runtime into
>
>
> >> "the total wall time $T_{\text{total}}$, the time devoted exclusively to the $\mu$ search $T_{\mu}$, the number of branch-and-bound nodes explored $N_{\text{nodes}}$, and the average time per node $T_{\text{node}} = T_{\mu} / N_{\text{nodes}}$." (Appendix D, p. 19)
>
> **These results show that, for fixed $K$ and dimension, the observed growth in runtime as $n$ increases matches the theoretical structure above: each node scales linearly in $|\hat S|$, and the number of nodes follows the reduced-space branching described in Q2**.
>
> ---
>
>
> > **Q4**. Would the method still converge on moderate hardware (e.g., 32 cores, 64 GB RAM)?
>
>
> **Yes. Convergence is a property of the algorithm and its branching scheme, not of the hardware. In Section 3.5, Theorem 3.5 states**:
>
> >> "Given an exhaustive subdivision on $\mu$, Algorithm 1 converges in the sense that $\lim_{i\to\infty} \alpha_i = \lim_{i\to\infty} \beta_i = z$."
>
> Algorithm 1 maintains a single global incumbent $\alpha_i$ and prunes any node $M$ with $\beta(M) \ge \alpha_i$. **Parallelization only distributes independent subproblems across cores**; it does not alter the subdivision of the centroid space, nor the bounding or pruning logic on which Theorem 3.5 relies. Consequently, the same convergence guarantee holds on more modest hardware (e.g., 32 cores, 64 GB RAM); the difference is purely in wall-clock time, not in whether the algorithm converges.
>
> > **Q5**. Can the same framework be extended to non-Euclidean or kernelized distances?
>
> **In its current form, SDC-GBB is developed and analyzed only for the Euclidean MSSC objective with squared Euclidean loss $|x_s - \mu_k|2^2$**. This choice is hard-wired in the MISOCP model in Section 2, eq. (2c), and in the two-stage reformulation where $Q_s(\mu) = \min{k \in K_s} |x_s - \mu_k|_2^2$ (Appendix A). All convergence and complexity results, as well as the geometric rules, are proved under this setting.
>
> **Several components of SDC-GBB essentially use Euclidean geometry**: the pseudo–sample collapse (Lemma 3.3, Theorem 3.4) relies on a Euclidean variance–decomposition identity, and the geometric sample-determination rules (Lemmas 3.1–3.2) and $d_{\min}/d_{\max}$ bounds assume explicit centroids $\mu_k \in \mathbb{R}^d$ and quadratic dependence on $\mu$. **In kernel k-means**, by contrast, centroids live implicitly in a reproducing kernel Hilbert space as combinations of feature maps, so there is no finite-dimensional centroid variable to branch on as in Algorithm 1, and the identities behind the collapse and pruning rules no longer hold in the same form.
>
> **Therefore, the paper does not claim an extension to non-Euclidean or kernelized distances**. Extending SDC-GBB would require re-deriving geometric bounds, pseudo-sample collapse, and lower-bounding schemes for the chosen dissimilarity or kernel, and proving new convergence and complexity guarantees—*this lies outside the scope of the present work and is a natural direction for future research.*

---

> ### Author Response · Authors · 2025-11-23
> **Follow-up – Does This Address Your Concerns?**
>
> Dear Reviewer Jhaj,
>
> As the discussion period is drawing to a close, we wanted to summarize our response regarding your concerns on novelty and scalability.
>
> **We clarified that our contribution restores decomposability to the constrained problem through exact ML collapse and geometric pruning**. This yields a complexity reduction from the cubic scaling of prior exact methods to linear scaling $O(|\hat{S}|K)$. This structural difference drives the $200\text{--}1500\times$ speedup and ensures convergence on standard resources, independent of hardware scale.
>
> *Do these clarifications resolve your doubts regarding the contribution and technical novelty? If not, please let us know so we can address any remaining points.*
>
> Best regards,
>
> Authors of Paper 14782

---

### Official Review · Reviewer_ja7J · 2025-10-31

**Soundness:** 2
**Presentation:** 2
**Contribution:** 2
**Rating:** 2
**Confidence:** 4

**Summary:**

This paper presents a new method to address the constrained clustering problem involving *cannot-link* (CL) and *must-link* (ML) constraints. The authors use a decomposable branch-and-bound (BB) framework that prunes cannot-link pairs using geometric rules. The used algorithm achieves highly scalable pairwise k-means constrained clustering through parallel computation.

**Strengths:**

1. The approach effectively combines mathematical rigor with computational efficiency, demonstrating potential scalability of the  MP(mathematical programming)-based method  to large datasets.

2. In general, the paper is easy to go through.

**Weaknesses:**

1. The proposed technique appears highly similar to [1], raising concerns about its originality and incremental contribution beyond existing work.

2. In prior research, must-link constraints can also be represented by representative points while preserving the problem's hardness.

**Questions:**

1. After introducing parallelism into the algorithm, how does the method maintain its optimality guarantees (as in the theoretical part)? Is the solution remains optimal? If not, what is the error bound?

2. Could the authors provide an analysis of the algorithm's running time complexity? Particularly, provide discussion of its scalability in terms of both data size and constraint density?

3.  Several critical questions for the experimental analysis:

a. Heuristic comparison:
   - Which specific heuristic methods were used in the experiments?
   - There exist many comparable approaches under the k-means setting (see Table 1 in [2]).
   - Could the authors compare against some of these methods and justify the selection of baselines?

b. Dataset consistency:
   - Tables 1, 2, and 3 appear to use different datasets. Could the authors provide complete results across all datasets for better comparison?

c. Table 4:
   - The authors describe certain observed phenomena without sufficient explanation.
   - Additionally, I believe TIME/NODE should be tested directly rather than reporting only *computing time per node*.

d. Table 5:
   - Why does the number of nodes decrease as the dataset size increases?

e. Fairness of comparison:
   - The experimental comparison appears potentially unfair.
   - For example, the paper compares BLPKM-CC and COP in terms of clustering cost, but metrics such as ARI, NMI, Purity, and running time are not reported. These should be included for completeness.

f. Table 6:
   - The results for SDC-GBB (CL) on the *iris* and *seeds* datasets remain identical across all metrics, suggesting the clustering assignments did not change.
   - Does this imply that the CL constraints have no influence (i.e., the solution remains unconstrained or was pruned by branch-and-bound)?
   - If so, in Table 8, why does BLPKM-CC achieve a lower cost than ENCODE k-means-post?
   - Generally, unconstrained costs are expected to be lower than constrained ones - please clarify this discrepancy.

[1] Kaixun Hua, Mingfei Shi, and Yankai Cao. *A scalable deterministic global optimization algorithm for clustering problems.* In *International Conference on Machine Learning (ICML)*, pp. 4391鈥?4401. PMLR, 2021.
[2] Philipp Baumann and Dorit S. Hochbaum. *An algorithm for clustering with confidence-based must-link and cannot-link constraints.* *INFORMS Journal on Computing*, 37(4):1044鈥?1068, 2025.

---

> ### Author Response · Authors · 2025-11-17
> **Response to Reviewer ja7J (1/5)**
>
> We appreciate the opportunity to further clarify our contributions. **The core scope of our work is to advance the scalability of deterministic global optimization for constraint clustering**, enabling high-quality, reproducible solutions on datasets orders of magnitude larger than previously possible by exact methods.
>
> > **W1.** The proposed technique appears highly similar to [1], raising concerns about its originality and incremental contribution beyond existing work.
>
> Our method is based on the reduced–space branch-and-bound (RS–BB) paradigm of [1], but [1] is designed for **unconstrained** MSSC and **cannot be applied directly** to pairwise constrained clustering.
>
> In [1], the two-stage extensive form is decomposable across samples: conditional on the centroids, each sample’s assignment subproblem is independent. This structural property is crucial for their Lagrangian decomposition and bounding strategy. In our setting, the hard pairwise constraints $(b_{s,k} = b_{s',k})$ and $(b_{s,k} + b_{s',k} \le 1)$ couple assignments across samples, so **the second-stage is no longer decomposable**. Simply adding (1b)–(1c) to [1] **breaks both** their LD-based lower bound and their upper-bounding scheme; **feasibility and global lower bounds are no longer guaranteed**.
>
> Our theoretical and algorithmic contribution is to restore decomposability and feasibility **exactly** in this constrained setting:
>
> * **Geometric sample determination with constraint propagation (Sec. 3.1).** We introduce new geometric rules that combine distance bounds with ML/CL propagation to fix or eliminate assignments while preserving feasibility at every node (Fig. 1).
> * **ML collapse equivalence (Sec. 3.2, Thm. 3.4).** We prove that collapsing each must-link component into repeated pseudo-samples yields an **unconstrained** MSSC instance whose global minimizer and objective value are identical to the original ML-only problem. **This is not a heuristic modeling trick**: it is a centroid-space equivalence that allows us to run RS–BB on the pseudo-sample instance **without losing global optimality** for the original problem.
> * **Constraint-aware bounds (Sec. 3.3–3.4).** We design a k-coloring-based upper bound for CL constraints and a grouped Lagrangian lower bound on the pseudo-sample instance. These bounds are new, explicitly incorporate pairwise constraints, and yield valid global lower bounds for the original constrained MSSC.
>
> On top of these theoretical and algorithmic extensions, we demonstrate deterministic $\varepsilon$-global optimization for constrained MSSC up to **1.5M** samples with must-link constraints and **200k** samples with cannot-link constraints—roughly **7× the largest unconstrained** instance treated in [1], and **200–1500× beyond existing exact constrained methods**.
>
> ---
>
> > **W2.** Must-link constraints have already been modeled via representative points in earlier work, so the novelty of your ML handling is unclear.
>
> **Our contribution is not using representatives**, but the exact integration of ML pseudosamples into a deterministic RS–BB framework **while preserving global optimality and scalability**.
>
> Concretely, in **Section 3.2 we prove** that collapsing each must-link component into repeated pseudosamples yields an unconstrained MSSC instance with identical global optimum to the original ML-only problem, and **Theorem 3.4 states that**
>
> >> *If $\mu^\*$ and $z(\mu^\*)$ are the global optimal solution and cost of Problem&nbsp;(4),  then they are also the global optimum and cost of the ML-only problem&nbsp;(2), and vice versa.*
>
> This is precisely what allows us to plug ML pseudosamples into the RS–BB scheme without losing any guarantees: all lower bounds, pruning rules and the convergence proof in **Section 3.3 and Appendix C** continue to apply to the aggregated instance, so the $\varepsilon$–global optimality certificate is preserved for the original ML-constrained MSSC.
>
> While prior super–points similar approaches either
> 1. are exact but assignment/SDP-based and remain limited to small n (Piccialli et al.), or
> 2. are heuristic Lloyd-type methods without global bounds (Baumann, BLPKM-CC),
>
> to our knowledge prior works that employ representative/super-point constructions **do not combine this kind of centroid-space equivalence with a deterministic RS-BB algorithm and certified global guarantees at the scales we demonstrate in Section 4.1**.

---

> ### Author Response · Authors · 2025-11-17
> **Response to Reviewer ja7J (2/5)**
>
> > **Q1.** After introducing parallelism into the algorithm, how does the method maintain its optimality guarantees
>
> **Does parallelism break optimality? No.** Theorem 3.5 (p. 5) states:
>
> > > *“Given an exhaustive subdivision on μ, Algorithm 1 converges in the sense that $\displaystyle \lim_{i\to\infty}\alpha_i = \lim_{i\to\infty}\beta_i = z$.”*
>
> The proof in **Appendix C** employs the convergence analysis of reduced-space BB from [8] and [3], which does not assume a single-threaded implementation. It assumes only (i) exhaustive subdivision and (ii) a bound-improving node selection rule; **neither depends on serial node processing.** Lemma C.1 and Lemma C.2 prove lower-bound consistency and convergence—**results that hold whether nodes are evaluated on one core or many**.
>
> **Is the solution still optimal? Yes—by construction.** Algorithm 1 maintains a **single global incumbent** $\alpha_i$ and prunes any node $M$ with $\beta(M) \ge \alpha_i$. Parallelization distributes the *evaluation* of nodes; the *logic* (bounding, pruning, incumbent update) is identical and synchronized. This is standard deterministic parallel BB, not heuristic search.
>
> In **Section 4.1 (Numerical Results, p. 6)**, we state that the relative optimality gap is computed in the same way in serial or parallel:
>
> > > *“the relative optimality gap is calculated as $\frac{\alpha_l-\beta_l}{\min(\alpha_l,\beta_l)}\times100%$”*. (Section 4.1, p. 6).
>
> **What is the error bound? The reported gap.** Every table in Section 4 shows the certified gap. When we stop at ≤0.1%, the solution is $ε$-optimal with $ε = 0.001$. If we hit the 12-hour limit, we report the final gap, **which remains a valid bound regardless of parallelism**.
>
> * [8] Cao, Y. and V. M. Zavala. “A scalable global optimization algorithm for stochastic nonlinear programs.”
> * [3] Hua, K., M. Shi, and Y. Cao. “A scalable deterministic global optimization algorithm for clustering problems.”
>
> ---
>
> > **Q2.** Could the authors provide an analysis of the algorithm's running time complexity? Particularly, provide discussion of its scalability in terms of both data size and constraint density?
>
> **The running-time complexity and scalability are analyzed in the manuscript; we briefly summarize here.** At each branch-and-bound node, the grouped Lagrangian bound and geometric filters cost $O(|\hat S|K)$, and exhaustive axis-aligned bisection yields at most $O((\delta/\varepsilon)^K m)$ leaves (Horst & Tuy, 2013). Hence the nominal worst-case complexity is $O((\delta/\varepsilon)^K m,|\hat S|K)$. Empirically, **Section 4.1 (Tables 1–3)** shows scalability from $n=150$ up to $n=1.5\times10^6$ (ML-only) and up to $n=2\times10^5$ (CL, ML+CL), while **Appendix D (Table 4)** studies constraint density: increasing ML density shrinks $|\hat S|$ and the number of nodes, and for CL constraints the number of nodes grows roughly linearly with the number of CL pairs, while the time per node remains essentially constant.
>
> For easy reference, we quote the relevant excerpts below:
>
> **Scalability in data size (Section 4.1, p. 7–8)**:
> >> “We evaluate the selected algorithms on 8 real-world datasets... Datasets are categorized as small (n ≤ 1,000), medium (n ≤ 10,000), large (n ≤ 100,000), and huge (n ≥ 100,000)... SDC-GBB successfully handles instances exceeding two hundred thousand samples across all constraint cases and further scales to ML-only instances with more than 1.5 million samples.”
>
> Tables 1–3 (pp. 7–9) report wall-clock time, node counts, and optimality gaps from $n=150$ to $n=1.5\times10^6$.
>
> **Scalability in constraint density (Appendix D, p. 17–18)**:
> >> “This section analyzes how varying densities of these constraints affect the branch-and-bound performance on medium-sized problem instances. Table 4 summarizes the number of nodes processed and the average computational time per node under different constraint densities... Under must-link constraints, SDC-GBB’s overall branching behavior varies substantially with constraint density... In contrast, the geometric sample-determination strategy for cannot-link constraints functions as barriers... the number of nodes explored grows roughly linearly under uniform cannot-link constraint placement; as we observe in Table 4, node count scales ∼O(m) with m cannot-link constraints, while time per node remains constant.”
>
> Table 4 (p. 18) provides explicit metrics for several constraint densities (from $n/2$ down to $n/64$).
>
> **Running-time complexity (Appendix D, p. 19)**:
> >> “When connecting this empirical measure to asymptotic complexity, we note that each branch-and-bound node incurs $O(|\hat S|K)$ work, while exhaustive axis-aligned bisection yields at most $O((\delta/\varepsilon)K m)$ (Horst & Tuy, 2013), where $\delta$ is the initial box size and $\varepsilon$ the target precision. Together these give a nominal worst-case cost of $O((\delta/\varepsilon)K m\,|\hat S|K)$.”

---

> ### Author Response · Authors · 2025-11-17
> **Response to Reviewer ja7J (3/5)**
>
> > **Q3.** a. Heuristic comparison: Which specific heuristic methods were used in the experiments? There exist many comparable approaches under the k-means setting (see Table 1 in [2]). Could the authors compare against some of these methods and justify the selection of baselines?
>
> **The heuristic methods, their roles, and the rationale for their selection are already specified in the manuscript**; we restate the key parts here.
>
> As stated in **Section 4 (Computational Experiments, p. 7)**:
>
> > > “(...) the best heuristic out of the following algorithms: **COP-k-means** (Wagstaff et al.), **encode-kmeans-post** (Nghiem et al.), **BLPKM-CC** (Baumann), and a **sensitivity-sampling coreset** algorithm (Feldman & Langberg). All heuristic algorithms were run with 100 restarts in 4 hours, and the full results are reported in Appendix E.”
>
> And in **Appendix E (Heuristic algorithms, pp. 19–20)**:
>
> > > “We evaluate the proposed procedure by comparing its clustering quality with four reference heuristics (...) subject to must-link (ML) and cannot-link (CL) constraints. **COP-k-means** (...) enforces the constraints at every assignment step. The post-processing **encode-k-means-post** (...) **respects all ML and CL relations**. The binary linear programming approach **BLPKM-CC** solves to optimality the assignment subproblem within each Lloyd iteration. Variants of coreset algorithm construct a ((k,\varepsilon))-coreset (...). Tables 7, 8, 9 report the optimal UB obtained by all heuristic algorithms with 100 independent initializations for ML-only, CL-only, and ML+CL experiments.”
>
> These methods were chosen because they represent **state-of-the-art heuristics for k-means with *hard* ML/CL constraints**, which is exactly our setting. Other approaches treat constraints as **soft penalties**, do not enforce strict feasibility, and therefore are not directly comparable to our exact constraint MSSC formulation.
>
> ---
>
> > **Q4.** b. Dataset consistency: Tables 1, 2, and 3 appear to use different datasets. Could the authors provide complete results across all datasets for better comparison?
>
> Tables 2 and 3 use a strict subset of the datasets in Table 1; this is intentional. The design is:
>
> * Table 1 (ML-only constraints) includes all datasets up to 1.5M, to showcase the scalability of SDC-GBB under must-link constraints, where ML components can be collapsed into pseudosamples.
> * Tables 2 and 3 (CL-only and ML+CL constraints) use the same subset of **12 datasets up to 210,000**. As discussed in the manuscript and its **Limitations (Section 6)**, instances with dense CL constraints are substantially harder; within a 12-hour time budget, it is unrealistic to solve 1M+–sample CL or ML+CL instances to small gaps.
>
> This is explained in the text (Section 4 and Section 6), but we agree it is useful to highlight it more explicitly.
>
> **For the camera-ready version**, we will add a footnote to Tables 2–3: *“Tables 2–3 use a subset of datasets from Table 1 (n ≤ 210,000). Table 1 includes larger instances (up to 1.5M samples) that are only tractable under ML constraints due to the scalability limitations discussed in Section 6.”*
>
> ---
>
> > **Q5.** c. Table 4: The authors describe certain phenomena without sufficient explanation. Additionally, I believe TIME/NODE should be tested directly rather than reporting only computing time per node.
>
> The phenomena in Table 4 are explained in **Appendix D**, which details sensitivity analysis on constraint density. Key excerpts:
>
> > > "Under must-link constraints, SDC-GBB's branching behavior varies with constraint density... the average time per node decreases with an increasing number of must-link constraints... there is a threshold of constraint density beyond which the branching process is significantly simplified. (...) In contrast, the geometric sample-determination strategy for cannot-link constraints functions as barriers... the number of nodes explored grows roughly linearly... while time per node remains constant." (Appendix D, pp. 19)
>
> In other words,
>
> * **ML constraints** (1): Merging reduces per-node work but adds equality constraints.
> * **CL constraints** (2): Geometric barriers increase node count, but early pruning keeps per-node time stable.
>
> This is further formalized by the **asymptotic complexity $O((δ/ε)^Km|\hat S|K)$** (Appendix D, pp. 19).
>
> **Regarding TIME/NODE**: This column **reports computational time per node**, measured empirically as total wall-clock time divided by nodes processed. We present this normalized metric (rather than raw times) to enable fair comparison across constraint densities.
>
> **For the camera-ready revision** we will: (i) Add a Table 4 footnote: *"TIME/NODE = wall-clock time divided by nodes processed"* and (ii) Update the caption to cross-reference Appendix D: *"See Appendix D for detailed analysis of these phenomena."*

---

> ### Author Response · Authors · 2025-11-17
> **Response to Reviewer ja7J (4/5)**
>
> > **Q6.** d. Table 5: Why does the number of nodes decrease as the dataset size increases?
>
> The trend in Table 5 — **fewer nodes for larger datasets** under fixed (n/4) ML density — is a direct consequence of **must-link component collapse**, as detailed in **Section 3.2 (pp. 4–5)** and **Appendix D (p. 17)**:
>
> >> “We show that collapsing each must-link component into repeated pseudo-samples yields an unconstrained instance with identical global optimum.” (Sec. 3.2)
>
> >> “Must-link constraints merge linked samples into single pseudo-points before initiating the branch-and-bound procedure… thus reducing the number of samples to consider.” (App. D)
>
> Under a fixed ML density of (n/4), increasing (n) in our synthetic construction **tends to produce larger connected components**, which collapse into **fewer pseudo-samples (|\hat S|)**. Empirically, this is exactly what happens in Table 5: the effective sample size (|\hat S|) reduces as (n) grows, and the branch-and-bound tree becomes shallower because it operates on a smaller aggregated instance.
>
> In the runtime analysis in **Appendix D.1**, we note that each node incurs (O(|\hat S|K)) work and the overall worst-case cost scales as
> [
> O\big((\delta/\varepsilon)^K m,|\hat S| K\big),
> ]
> so reducing (|\hat S|) effectively reduces the search effort on the ML-aggregated instance.
>
> **For the camera-ready revision**, we will add a footnote to Table 5: *“Node count decreases with (n) because larger ML components reduce (|\hat S|) via pseudo-sample collapse (Section 3.2), enabling scalability up to 1.5M samples under ML constraints.”*
>
> ---
>
> > **Q7**. e. Fairness of comparison: The experimental comparison appears potentially unfair. For example, the paper compares BLPKM-CC and COP in terms of clustering cost, but metrics such as ARI, NMI, Purity, and running time are not reported. These should be included for completeness.
>
> Our experimental design follows a **standard distinction in the clustering literature** ([1,2]), which we explicitly state in **Appendix D.2 (Clustering Evaluation)**:
>
> >> “Our work distinguishes between two aspects of clustering performance: (i) the formulation, which defines how data similarity is measured and affects metrics such as ARI, NMI, and purity; and (ii) the quality of the solution (in terms of cost minimization), which we measure using the optimality gap.” (App. D.2)
>
> Our primary contribution is a deterministic global optimizer for MSSC under **hard ML/CL constraints**. Consequently, **SSE and its optimality gap** are the natural and rigorous measures of solution quality — especially given that we provide **certified global bounds**, which existing constrained methods do not offer at comparable scales.
>
> That said, we **do report ARI, NMI, and Purity** on labeled datasets (IRIS, SEEDS, HEMI, HTRU2, SKIN_8) in **Table 6 (p. 20)**, showing that SDC-GBB matches or exceeds the unconstrained MSSC baseline of Hua et al. (2021) even under strict ML/CL constraints. This supports that our globally optimal SSE solutions correspond to meaningful cluster structure.
>
> For **BLPKM-CC** and **COP-k-means** in Tables 7–9, these baselines are **secondary** and are used solely to obtain heuristic upper bounds. All of them optimize the **same SSE objective under identical hard constraints**, so **SSE alone** is the relevant and fair comparison metric in this context. Reporting ARI/NMI here would not add information about progress toward the global constrained optimum and is potentially misleading, since these external indices are sensitive to label permutations and to formulation choices rather than to optimality gaps.
>
> Regarding **running time**, the complexity and runtime behavior of SDC-GBB are analyzed in **Appendix D.1** via detailed wall-clock, time-per-node, and core-hour statistics (Table 5). All heuristic methods (COP-k-means, ENCODE k-means-post, BLPKM-CC, Sensitivity Sampling) are run under the **same 4-hour cutoff**, and in practice they complete strictly within this budget; their role is to provide reference upper bounds, not competing global guarantees.
>
> **For the camera-ready revision**, we will (i) add a one-sentence summary to Section 5.1 emphasizing that **SSE optimality gap is our primary evaluation metric** for solution quality. and (ii) expand the Table 6 caption to clarify that **ARI/NMI/Purity are reported only as external validation**, while **solution quality is measured via certified SSE gaps**.
>
> * [1] Ben-David & Ackerman, “Measures of Clustering Quality: A Working Set of Axioms for Clustering”, NeurIPS 2008.
> * [2] Jain, “Data Clustering: 50 Years Beyond k-Means”, Pattern Recognition Letters 2010.
> * [3] Hua et al., “A Scalable Deterministic Global Optimization Algorithm for Clustering Problems”, ICML 2021.

---

> ### Author Response · Authors · 2025-11-17
> **Response to Reviewer ja7J (5/5)**
>
> > **Q8.** f. Table 6: The results for SDC-GBB (CL) on the iris and seeds datasets remain identical across all metrics, suggesting the clustering assignments did not change. Does this imply that the CL constraints have no influence (i.e., the solution remains unconstrained or was pruned by branch-and-bound)?
>
> The fact that the results for SDC-GBB (CL) on Iris and Seeds coincide with the unconstrained solution **does not mean that CL constraints are ignored**. As documented in the paper (Our Contributions, p. 2):
>
> >> “We devise geometric sample-determination rules that eliminate cannot-links, which specify whether points must not be placed into the same clusters before enumeration.”
>
> These rules are formalized in **Lemmas 3.1–3.2** (Section 3.1, p. 3). **Algorithm 1** (Section 3.5, p. 6) enforces them at every node:
>
> >> Propagate fixes via $\mathcal{T}^{\mathrm{ml}}$, $\mathcal{T}^{\mathrm{cl}}$; update $K_s$.  (Section 3.5, p. 6)
>
> Here, $\mathcal{T}_{\mathrm{cl}}$ is the cannot-link set. This feasibility check runs at every branch-and-bound node.
>
> **Why the metrics are identical:** on small, well-separated datasets like Iris and Seeds, the **unconstrained global optimum happens to satisfy all randomly generated CL pairs**. The constraints are **active but non-binding**: the algorithm still validates every CL pair through Lemmas 3.1–3.2 and node-level feasibility checks, but no additional pruning is triggered because the optimal solution already complies.
>
> **For the camera-ready revision**, we will add a footnote to Table 6: *“Identical metrics confirm that CL constraints are satisfied at the global optimum; enforcement via Lemmas 3.1–3.2 and node feasibility checks remains active throughout the tree.”*
>
> ---
>
> > **Q9.** If so, in Table 8, why does BLPKM-CC achieve a lower cost than ENCODE k-means-post? Generally, unconstrained costs are expected to be lower than constrained ones – please clarify this discrepancy.
>
> **There is no unconstrained method in Table 8.** As stated in **Appendix E (p. 20)**:
>
> * ENCODE k-means-post [4]:
>
>   > “formulates the reassignment … as a binary combinatorial program that respects all ML and CL relations.”
> * BLPKM-CC [5]:
>
>   > “solves to optimality the assignment subproblem within each Lloyd iteration under the same constraints.”
> * COP-k-means [6]:
>
>   > “enforces the constraints at every assignment step.”
>
> Thus, **all baselines in Tables 7–9 are constrained methods** solving the **same ML/CL-constrained MSSC**.
>
> The expectation “unconstrained SSE $\le$ constrained SSE” holds **only at global optima, not at local minima**. All methods in Table 8 are Lloyd-type heuristics that converge to **different constrained local minima**, so there is no reason to expect a monotonic ordering among their final SSE values.
>
> Lloyd’s algorithm (which underlies these heuristics) is known to converge to a local optimum and can be **arbitrarily bad in theory** [7]. The methods in Table 8 thus end up in different constrained local minima. It is therefore entirely expected—and observed on Seeds—that **BLPKM-CC reaches a better constrained local minimum than ENCODE**, even though both satisfy the same CL constraints.
>
> This behavior is a **standard consequence of the non-convex MSSC objective** and the well-known sensitivity of Lloyd-style algorithms to initialization and search path. By contrast, **our method certifies the global constrained optimum**, eliminating local-minimum ambiguities.
>
> * [4] Nghiem, Nguyen-Viet-Dung, et al. “Constrained clustering via post-processing.” Discovery Science, 2020.
> * [5] Baumann, Philipp. “A binary linear programming-based k-means algorithm for clustering with must-link and cannot-link constraints.” IEEM, 2020.
> * [6] Wagstaff et al. “Constrained k-means clustering with background knowledge.” ICML, 2001.
> * [7] Vattani, Andrea. “K-means requires exponentially many iterations even in the plane.” SoCG, 2009.

---

> ### Author Response · Authors · 2025-11-23
> **Follow-up – Does This Address Your Concerns?**
>
> Dear Reviewer ja7J,
>
> As the discussion period is drawing to a close, **we wanted to check if our detailed responses have addressed your concerns** regarding the novelty of our approach (*specifically the exact ML collapse and geometric pruning*) and the source of our scalability (*algorithmic decomposability vs. hardware*). We also provided clarifications on the experimental results in Tables 4–6 and the fairness of our baselines.
>
> *Do these clarifications resolve your doubts regarding the contribution and technical novelty? If not, please let us know so we can address any remaining points.*
>
> Best regards,
>
> Authors of Paper 14782

---

> > ### Comment · Reviewer_ja7J · 2025-11-28
> >
> > Thanks for the reply. It has partially cleared up my confusion. I will adjust my score accordingly. My biggest concern remains that the theoretical contribution of the paper is marginal.

---

> ### Author Response · Authors · 2025-11-28
>
> Thank you for the efforts spent on reviewing our paper.
>
> The main theoretical contribution of our work is to provide **the first decomposable global optimization framework for pairwise constrained k means**.
>
> Our approach
>
> (i) proves an exact must link collapse equivalence that maps a problem with must link constraints to an unconstrained minimum sum of squares clustering problem with the same global optimum,
>
> (ii) derives geometric rules that propagate must link and cannot link constraints while preserving feasibility of all nodes in the search, and
>
> (iii) combines these elements in a reduced space branch and bound algorithm with grouped sample Lagrangian bounds that produces **certified global solutions for data sets with up to 1.5 million ML and 200,000 CL samples**.
>
> **For the camera-ready revision**, we will revise the introduction and the theory section in order to state these contributions clearly and to explain how they support the empirical scalability of the method.
>
> *Please let us know if any concern remains.*

---

### Meta-Review · Area_Chair_FeaQ · 2026-01-05

**Summary:**

This paper proposes a scalable deterministic global optimization algorithm for the minimum sum-of-squared clustering (MSSC) task with pairwise ML and CL constraints. The authors introduce a centroid-based pseudosample formulation for must-link subsets, leveraging the combined information of each group to maintain the exact global minimum while reducing problem complexity. Finally,  they provide a branch-and-bound algorithm.

After reading the comments from four reviewers, together with my own assessment, I think there are three major issues for this manuscript:

1) The novelty comparing with previous work (like the prior branch-and-bound methods and constrained clustering methods). Although the authors explained the differences in their rebuttal, the emphasized differences mainly lie on the heuristics, not fundamental or theoretical aspects.

2) Experiments. I do think the current experimental results are not sufficient. First, all the studied datasets are low dimensional (less than 10). Second, the constructed synthetic datasets are very simple, which are generated by Gaussian or random. Also, another important parameter "k" is fixed to be 3 in Tab 1-3.

3) The presentation needs to improve. For example, I suggest the authors to clearly state some key parts of the algorithm, like the time complexity.

**Reviewer Concerns:**

The reviewers raised several concerns on the novelty, experiments, and presentation. I appreciate the authors for providing detailed explanation on some questions. However, I think there is still a large room to improve for this manuscript.

**Reviewer Scores:**

The reviewers initially give 2, 4, 4, 4. Two of the reviewers want to increase their scores, but the overall assessment is still negative.

---

### Decision · Program_Chairs · 2026-01-26

Reject